# S-acylation of NLRP3 provides a nigericin sensitive gating mechanism that controls access to the Golgi

**Daniel M Williams\*, Andrew A Peden\***

School of Bioscience, University of Sheffield, Sheffield, United Kingdom

## eLife assessment

This **important** paper implicates S-acylation of Cys-130 in recruitment of the inflammasome receptor NLRP3 to the Golgi, and it provides **convincing** evidence that S-acylation plays a key role in response to the stress induced by nigericin treatment. While Cys-130 does seem to play a previously unappreciated role in membrane association of NLRP3, further work will be needed to clarify the details of the mechanism.

**\*For correspondence:**
danielmwilliams12@gmail.com (DMW);
a.peden@sheffield.ac.uk (AAP)

**Competing interest:** The authors declare that no competing interests exist.

**Abstract** NLRP3 is an inflammasome seeding pattern recognition receptor activated in response to multiple danger signals which perturb intracellular homeostasis. Electrostatic interactions between the NLRP3 polybasic (PB) region and negatively charged lipids on the trans-Golgi network (TGN) have been proposed to recruit NLRP3 to the TGN. In this study, we demonstrate that membrane association of NLRP3 is critically dependant on S-acylation of a highly conserved cysteine residue (Cys-130), which traps NLRP3 in a dynamic S-acylation cycle at the Golgi, and a series of hydrophobic residues preceding Cys-130 which act in conjunction with the PB region to facilitate Cys-130 dependent Golgi enrichment. Due to segregation from Golgi localised thioesterase enzymes caused by a nigericin induced breakdown in Golgi organisation and function, NLRP3 becomes immobilised on the Golgi through reduced de-acylation of its Cys-130 lipid anchor, suggesting that disruptions in Golgi homeostasis are conveyed to NLRP3 through its acylation state. Thus, our work defines a nigericin sensitive S-acylation cycle that gates access of NLRP3 to the Golgi.

## Introduction

Inflammasomes are multimolecular complexes that control the initiation of inflammatory responses to various damage associated molecular patterns (DAMPs) and pathogen associated molecular patterns (PAMPs; *Chou et al., 2023*). The core complex involved in the formation of an active inflammasome consists of a pattern recognition receptor (PRR), which sense various perturbations in intracellular homeostasis and are the trigger point for inflammasome activation, the adaptor protein ASC, which acts to amplify recognition of DAMPs and PAMPs by PRRs and ensure a robust and rapid response, and a final effector protein – Caspase-1, which, amongst other substrates, cleaves the pro-inflammatory cytokine interleukin-1-beta (IL-1β) and the pore forming protein Gasdermin-D into their bioactive forms. Due to its involvement in the pathogenesis of several diseases, including atherosclerosis (*Duewell et al., 2010*) and Alzheimer's disease (*Heneka et al., 2013*), one of the most well-studied inflammasome seeding PRRs is NLRP3 (**N**ACHT, **L**RR and **P**YD domain containing protein **3**; *Akbal et al., 2022*; *Fu and Wu, 2023*; *Liang et al., 2021*; *Seoane et al., 2020*; *Swanson et al., 2019*). Stimuli that trigger canonical NLRP3 activation are diverse,

ranging from bacterial ionophores such as nigericin (*Perregaux et al., 1992*) to agents which damage lysosomal integrity (*Duewell et al., 2010*; *Hornung et al., 2008*). The majority of these stimuli are unified by their ability to deplete intracellular potassium, which has been proposed to be a common danger signal sensed by NLRP3 (*Muñoz-Planillo et al., 2013*). Recent studies have revealed that NLRP3 is a peripheral membrane protein, recruited to the Golgi apparatus and endosomes through interactions between a highly conserved polybasic (PB) region within NLRP3 itself, and negatively charged phospholipids such as phosphatidylinositol-4-phosphate (PI4P) found in the cytoplasmic leaflet of intracellular membranes (*Chen and Chen, 2018*; *Lee et al., 2023*; *Schmacke et al., 2022*; *Zhang et al., 2023*). Membrane binding is important for formation of an inactive NLRP3 oligomer and an optimal response to NLRP3 stimuli (*Andreeva et al., 2021*; *Chen and Chen, 2018*), although how membrane binding and sensing of NLRP3 stimuli are linked is at present unclear.

The presence of a PB motif alone is often insufficient to allow stable membrane association. As one example directly relevant to NLRP3, basic residues within the hyper-variable region (HVR) at the C terminus of K-Ras4B, a membrane-bound GTPase that mediates key signalling events from the plasma membrane (*Zhou et al., 2018*), are paired with a proximal lipid anchor to confer specific localisation to the plasma membrane, with neither module alone sufficient for plasma membrane binding (*Choy et al., 1999*; *Hancock et al., 1991*; *Hancock et al., 1990*). A working model for the steady state localisation of K-Ras4B at the plasma membrane posits that the farnesyl group provides an initial non-specific weak affinity for all intracellular cellular membranes. The PB region acts in concert with the farnesyl lipid anchor to allow K-Ras4B to preferentially partition to the plasma membrane through complementary charge-based interactions between the PB region and the high net negative charge of the plasma membrane imparted by the relative abundance of several species of negatively charged lipids (*Schmick et al., 2014*; *Yeung et al., 2008*; *Zhou et al., 2017*).

The 'two signal' model of stable association of peripheral membrane proteins lacking folded lipid binding domains with intracellular membranes also applies to additional members of the Ras family and many other peripheral membrane proteins (*Hancock et al., 1989*; *Heo et al., 2006*; *Laude and Prior, 2008*; *Linder et al., 1993*; *Michaelson et al., 2001*). In the absence of a PB region, a second membrane binding module can be provided by an additional lipid anchor, such as the covalent attachment of an acyl group to free cysteine residues. For the farnesylated forms of H-Ras and N-Ras, Golgi localised palmitoyl-acyltransferases (PATs) catalyse the addition of an acyl group to cysteine residues adjacent to the farnesylation motif at the C terminus of H-Ras and N-Ras, trapping these proteins at the Golgi (*Hancock et al., 1989*; *Rocks et al., 2010*; *Rocks et al., 2005*; *Swarthout et al., 2005*). The opposing action of thioesterase enzymes, such as those of the APT and ABHD families (*Anwar and van der Goot, 2023*; *Won et al., 2018*), to remove covalently attached acyl groups helps to prevent the equilibration of S-acylated peripheral membrane proteins across all intracellular membranes through vesicular transport, concentrating H-Ras and N-Ras at the Golgi and downstream compartments such as the plasma membrane (*Dekker et al., 2010*; *Goodwin et al., 2005*; *Rocks et al., 2010*; *Rocks et al., 2005*; *Salaun et al., 2010*).

At present it is unclear how NLRP3 preferentially associates with the Golgi given that only a PB region in NLRP3 has been described (*Chen and Chen, 2018*). In this study, we went in search of additional signals in NLRP3 that may mediate recruitment of NLRP3 to intracellular membranes alongside the PB region. We find that the Golgi association of NLRP3 is dependent on S-acylation of NLRP3 at a highly conserved cysteine residue (Cys-130) adjacent to the PB region with NLRP3 dynamically associating with the Golgi through reversible S-acylation at Cys-130. We also identify a string of hydrophobic residues upstream of Cys-130 that are also important for recruitment of NLRP3 to the Golgi. Enhanced Golgi recruitment of NLRP3 in response to nigericin is likely caused by alterations in the NLRP3 S-acylation cycle, which leads to a fraction of NLRP3 becoming immobilised on the Golgi. We present data consistent with this immobilisation being attributable to nigericin induced alterations in Golgi organisation and function, which potentially limits the ability of Golgi resident thioesterase enzymes to encounter NLRP3 and de-acylate it. Overall, our results provide an enhanced understanding of the features within NLRP3 required for membrane binding and suggest that disruptions in Golgi homeostasis are coupled to alterations in the NLRP3 S-acylation cycle.

## Results

### The NLRP3 polybasic region is insufficient to localise NLRP3 to the Golgi

Previous work on NLRP3 has demonstrated that association with TGN46 positive structures is dependent on the NLRP3 polybasic region between residues 131–145. To determine if the NLRP3 polybasic region alone was sufficient to mediate Golgi localisation, we fused the NLRP3 polybasic sequence to GFP (GFP-NLRP3[131-158]) and expressed this construct in HeLaM cells (*Figure 1A*). Expression of GFP-NLRP3[131-158] was insufficient to achieve a Golgi localisation however and predominantly localised to the nucleus (*Figure 1B* - top row, 3rd panel). As polybasic regions are often paired with a 'second signal' to confer a specific localisation (*Heo et al., 2006*), we next tested whether the addition of a second membrane binding signal was sufficient to allow association of GFP-NLRP3[131-158] with intracellular membranes. For this purpose, we appended a C terminal farnesylation motif (CVIM) to GFP-NLRP3[131-158] (GFP-NLRP3[131-158-CVIM]; *Figure 1A*). Similar to the charge biosensor probe +8 Pre (*Eisenberg et al., 2021*; *Roy et al., 2000*; *Yeung et al., 2008*), which is farnesylated and contains a PB region with a similar net positive charge to the NLRP3 PB region, expression of GFP-NLRP3[131-158-CVIM] resulted in strong recruitment of GFP to the plasma membrane (*Figure 1B* - middle row, 2nd and 3rd panels, quantified in *Figure 1C*). Thus, whilst the PB region alone is insufficient to drive association with intracellular membranes, residues 131–158 within NLRP3 can act in conjunction with a second membrane binding signal to confer binding to specific intracellular membranes.

### Identification of additional features necessary for NLRP3 Golgi localisation

Dual membrane binding signals in peripheral membrane proteins are often located near one another in a proteins primary sequence (*Hancock et al., 1989*; *Hancock et al., 1990*; *Heo et al., 2006*). As the PB region alone was insufficient for membrane recruitment, we therefore examined the NLRP3 sequence around the PB region for additional features that may contribute to recruitment of NLRP3 to the Golgi. Two features preceding the NLRP3 polybasic region stand out; a highly conserved cysteine residue at position 130 (Cys-130), which we hypothesised may be S-acylated, and a series of hydrophobic residues upstream of Cys-130 (*Figure 1D*). We speculated that these hydrophobic residues may fulfil an analogous role to the farnesyl anchor in N-Ras and H-Ras, providing a weak non-specific membrane affinity. In a recently published Cryo-EM structure of the inactive NLRP3 complex bound to CRID3 (*Hochheiser et al., 2022*), the hydrophobic residues form part an alpha helix (residues 115–125, helix[115-125]), with several hydrophobic residues (I113, W117, L120, L121, L24) aligned along a single face (*Figure 1D and E*, *Figure 1—figure supplement 1A–C*). Cys-130 resides between helix[115-125] and the polybasic region. Helix[115-125] and Cys-130 sit at the centre of one NLRP3 pentamer and are largely surface exposed, as they face away from the core of the decameric complex (*Figure 1—figure supplement 1D*), suggesting these additional features may work together with the polybasic region to allow localisation to the Golgi.

To first determine if NLRP3 Golgi association was dependent on S-acylation, we treated cells with 2-bromopalmitate (2 BP), a palmitic acid analogue which is thought to block global protein S-acylation through covalent binding to the active site of ZDHHC enzymes (*Davda et al., 2013*; *Jennings et al., 2009*; *Webb et al., 2000*). Golgi enrichment of NLRP3 in HeLaM cells transiently transfected with GFP-tagged NLRP3 was inhibited by treatment of cells with 2 BP (*Figure 1F and G*), suggesting that S-acylation regulates recruitment of NLRP3 to the Golgi. 2 BP induced loss of NLRP3 from the Golgi was not due to alterations in Golgi PI4P levels, which remained similar in both untreated and 2BP-treated cells (*Figure 1—figure supplement 2A*), or due to an overall loss of Golgi integrity (*Figure 1—figure supplement 2B–E*).

Several studies have demonstrated that NLRP3 stimuli cause an increase in the recruitment of NLRP3 to intracellular membranes, which have been proposed to represent either the Golgi (*Chen and Chen, 2018*) or endosomes (*Lee et al., 2023*; *Zhang et al., 2023*). Consistent with these studies, we observed a similar increase in NLRP3 signal within the perinuclear area following nigericin treatment (*Figure 1—figure supplement 3A–C*). The membranes NLRP3 resides on post-stimulation are likely derived from the Golgi as both GFP-NLRP3 (*Figure 1—figure supplement 4A–C*) and an untagged-NLRP3 construct (*Figure 1—figure supplement 4D*) were sensitive to BrefeldinA (BFA),



**Figure 1.** Association of NLRP3 with intracellular membranes is dependent on S-acylation. (**A**) Schematic of the sequences used for GFP tagged minimal NLRP3 constructs expressed transiently in HeLaM cells. (**B**) Representative confocal microscopy images of each construct expressed. Scale bar = 10 µm. (**C**) Quantification of the amount of each mini-construct localised to the Golgi relative to cytosolic signal. N=3. (**D**) Overview of domain architecture of NLRP3. Boxed region indicates region and residues shown below with sequence conservation across 20 species represented by a web-logo plot. (**E**) ChimeraX graphic based on the inactive Cryo-EM structure of NLRP3 (PDB ID: 7PZC) showing a side view of the NLRP3 decamer. Monomers within each pentamer are colour coded blue or green. One monomer from each pentamer is hidden to give a clearer view of helix[115-125] (coloured cyan with Cys-130 in pink) and the polybasic region (coloured yellow) from each monomer. (**Ei**) Graphic from boxed region showing the position of residues 95–146 relative to the PYD and NACHT domains. (**Eii**) Zoom in looking along helix[115-125]. Residues Tryp117, Leu120, Leu121, and Leu124 (labelled) align along a single face of helix[115-125]. (**F**) Representative confocal microscopy images of HeLaM cells transiently expressing GFP-NLRP3 treated with either 0.5% DMSO or 100 µM 2 BP for 16 hr with or without a 1 hr treatment with 10 µM nigericin and (**G**) quantification of GFP-NLRP3 signal associated with the Golgi from the same experiment. N=4. Scale bars = 10 µm. (**H**) Representative images of HeLaM cells transiently expressing GFP-NLRP3 or GFP-NLRP3[C130S] treated with or without 10 µM nigericin for 1 hr. Scale bars = 10 µm. (**I**) Quantification of GFP-NLRP3 or GFP-NLRP3[C130S] on the Golgi before and after nigericin treatment. N=3.

The online version of this article includes the following figure supplement(s) for figure 1:

**Figure supplement 1.** Cysteine-130 and helix[115-125] are surface exposed in the cryo-EM structure of the inactive NLRP3 complex.

**Figure supplement 2.** Treatment of cells with 2 BP does not impact Golgi PI4P levels or overall Golgi integrity.

**Figure supplement 3.** Dynamics of NLRP3 association with the Golgi post nigericin treatment in HeLaM cells.

**Figure supplement 4.** NLRP3 localisation is sensitive to BrefeldinA.

*Figure 1 continued on next page*

*Figure 1 continued*

**Figure supplement 5.** Localisation of untagged NLRP3 to the Golgi is S-acylation dependent.

**Figure supplement 6.** Hydrophobic residues within helix[115-125] are important for localisation of NLRP3 to the Golgi apparatus.

**Figure supplement 7.** Farnesylation of NLRP3 residues 95–158 permits Cys-130-dependent Golgi association.

**Figure supplement 8.** Definition of a minimal NLRP3 region required for enrichment at the Golgi apparatus.

a Golgi disassembling fungal metabolite that has previously been shown to inhibit NLRP3 activation (*Hong et al., 2019*; *Phulphagar et al., 2021*; *Zhang et al., 2017*). BFA treatment abolished Golgi enrichment of NLRP3 at steady state and prevented accumulation of NLRP3 in the perinuclear region following nigericin treatment (*Figure 1—figure supplement 4A–D*). To test whether enhanced recruitment of NLRP3 to the Golgi in response to nigericin was dependent on S-acylation, we pre-treated cells with 2 BP overnight prior to a 1 hr stimulation with nigericin. 2BP treatment prevented the perinuclear clustering of NLRP3 signal observed following nigericin stimulation and instead induced the formation of peripherally localised aggregates (*Figure 1F and G*), indicating that enhanced recruitment of NLRP3 to the Golgi in response to NLRP3 stimuli is S-acylation dependent. The puncta seen following nigericin stimulation of 2BP-treated cells showed minimal overlap with the early endosomal marker EEA1 (*Figure 1—figure supplement 2E, F*) or PI4P (*Figure 1—figure supplement 2A*), indicating that these structures do not represent recruitment of NLRP3 to early endosomes. In agreement with Cys-130 being a critical determinant of NLRP3 Golgi localisation, mutation of this residue phenocopied the effect of 2 BP, preventing Golgi enrichment of GFP-NLRP3 at steady state and in response to nigericin treatment (*Figure 1H, I*). The sensitivity of NLRP3 membrane binding to 2 BP and mutation of Cys-130 could also be seen with untagged NLRP3, although untagged NLRP3[C130S] did not form peripheral puncta in response to nigericin, suggesting these may be an artifact of GFP tagging (*Figure 1—figure supplement 5A*).

To test whether the hydrophobic residues within helix[115-125] are also important for membrane binding we mutated residues isoleucine-113, tryptophan-117, leucine-120, and leucine-124 (hydrophobic face mutant; NLRP3[HF]), which are predicted to form a hydrophobic face as part of helix[115-125] (*Figure 1—figure supplement 6A, B*). These mutations abolished recruitment of NLRP3 to the Golgi at steady state and in response to nigericin (*Figure 1—figure supplement 6C, D*), indicating that these residues also play an important role in recruitment of NLRP3 to intracellular membranes alongside Cys-130 and the polybasic region.

## Definition of a minimal NLRP3 membrane binding region

To determine if Cys-130 and helix[115-125] were sufficient in isolation to drive Golgi localisation, we again expressed minimal NLRP3 peptides containing these residues in cells. GFP alone localised to both the nucleus and cytoplasm (*Figure 1B*, top row, 1st panel). In contrast, expression of residues 95–130 (NLRP3[95-130]), containing both helix[115-125] and Cys-130, localised predominantly to the cytoplasm with some Golgi enrichment visible, however this was not at levels similar to full length NLRP3 (*Figure 1B* – bottom row, 1st panel). Surprisingly, inclusion of the polybasic region (NLRP3[95-158]) also failed to recapitulate Golgi enrichment equivalent to the full-length protein (*Figure 1B* – bottom row, 2nd panel), suggesting that additional features within NLRP3 are necessary to achieve recruitment to the Golgi. Consistent with the need for an additional membrane affinity for maximal binding of NLRP3 to the Golgi, attachment of a C-terminal farnesylation sequence (CVIM) to NLRP3[95-158] (NLRP3[95-158-CVIM]) was sufficient to achieve levels of Cys-130 dependent Golgi enrichment comparable to full length NLRP3 (*Figure 1B*, bottom row, 3d panel, quantified in *Figure 1C*) with this construct also showing a high degree of overlap with NLRP3 at the Golgi (*Figure 1—figure supplement 7A–E*). Thus, whilst helix[115-125], Cys-130 and the polybasic region are all essential for Golgi localisation, additional properties in NLRP3 are required to allow full Golgi binding.

To attempt to define a minimal NLRP3 region that shows levels of Golgi enrichment equivalent to the full-length protein, we created NLRP3 mutants truncated from either the N or C terminus. Removal of the pyrin domain (PYD, residues 1–95) had no impact on Golgi localisation; however, further truncation to residue 121 prevented association of NLRP3 with the Golgi in both untreated and nigericin treated cells, consistent with the requirement for an intact helix[115-125] (*Figure 1—figure supplement 8A–C, H*). A C-terminal NLRP3 truncation lacking the LRR domain (NLRP3[1-699]) localised

to the Golgi at reduced levels, with further truncation down to residue 680 largely abolishing NLRP3 Golgi enrichment under both control and nigericin treated conditions (*Figure 1—figure supplement 8D–F, H*). Expression of a minimal NLRP3 construct spanning residues 95–699 (NLRP3[95-699]) behaved the same as NLRP3[1-699], confirming that the LRR domain is important for optimal association of NLRP3 with the Golgi and that the PYD domain is dispensable (*Figure 1—figure supplement 8G, H*). Collectively, these results indicate that the presence of the LRR domain boosts NLRP3 membrane affinity, potentially through its role in establishing the formation of a multimeric NLRP3 complex (*Andreeva et al., 2021*; *Hochheiser et al., 2022*).

## ZDHHC3 and ZDHHC7 overexpression enhance the Golgi localisation of NLRP3

Protein S-acylation is mediated by a family of 23 palmitoyltransferases (PAT) in humans (*Lemonidis et al., 2015*). PATs are multi-pass transmembrane proteins defined by the presence of a characteristic zinc dependent DHHC domain (zDHHC) responsible for the covalent attachment of acyl chains of varying length to exposed cysteine residues on target substrates (*Fukata et al., 2004*; *Greaves*

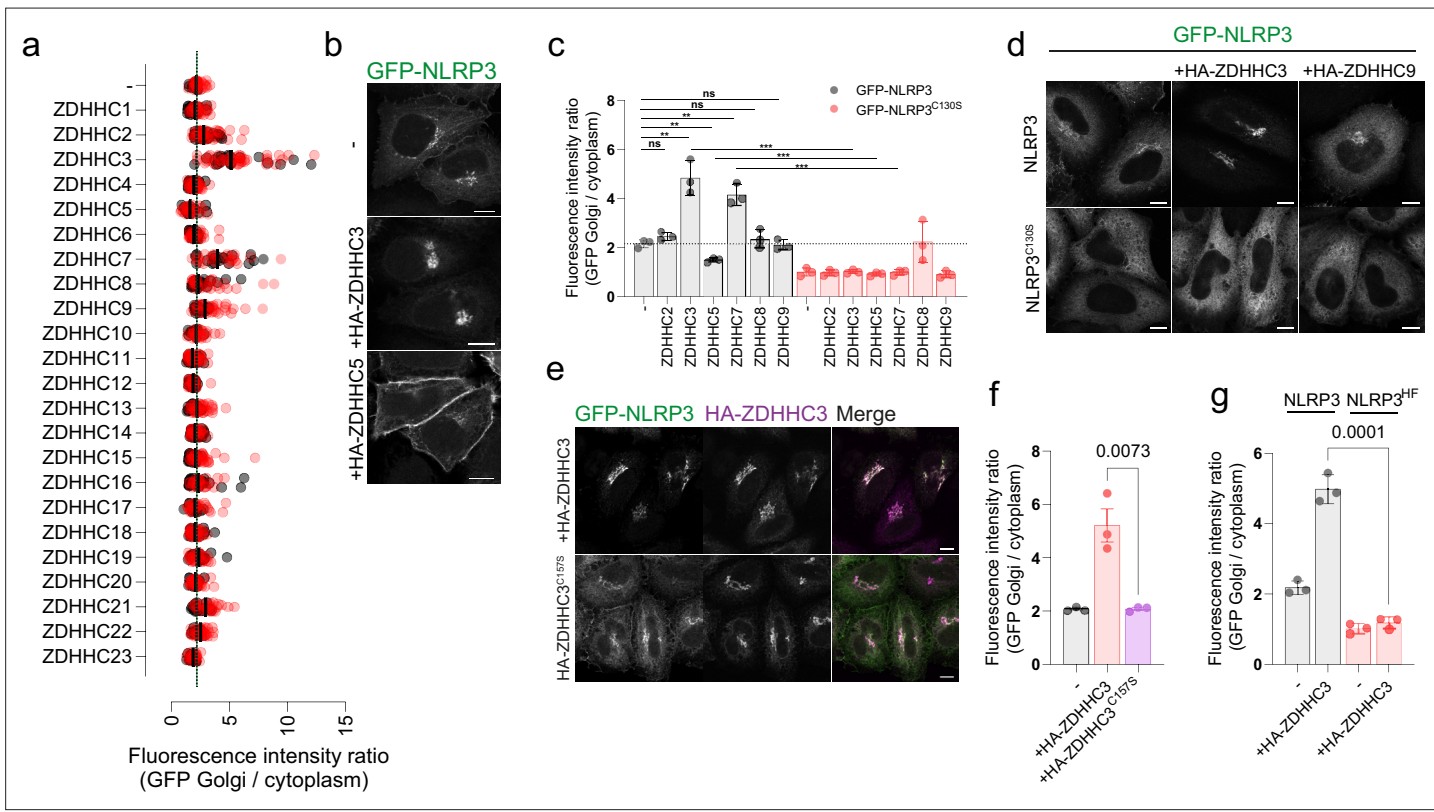

**Figure 2.** Overexpression of ZDHHC palmitoyltransferases can alter NLRP3 localisation in a Cys-130 and helix[115-125]-dependent manner. (**A**) Quantification of the ratio of GFP-NLRP3 signal on the Golgi relative to that in the cytosol in HeLaM cells transiently transfected with 1 μg of each individual HA-tagged ZDHHC enzyme and GFP-NLRP3. N=2. (**B**) Representative confocal microscopy images showing the effect of HA-ZDHHC3 and HA-ZDHHC5 on the localisation of GFP-NLRP3. Scale bars = 10 μm. (**C**) Quantification of GFP-NLRP3 or GFP-NLRP3[C130S] Golgi signal in HeLaM cells transiently transfected with lower amounts (0.25 μg) of each HA-ZDHHC enzyme which altered GFP-NLRP3 localisation in (**A**). N=3. **p<0.01, ***p<0.001 (Unpaired t test). (**D**) Representative confocal microscopy images of GFP-NLRP3 or GFP-NLRP3[C130S] expressed with the indicated HA tagged ZDHHC enzyme. HA-ZDHHC9 images demonstrate that not all Golgi localised PATs drive enhanced Golgi recruitment of NLRP3. Scale bars = 10 μm. (**E**) Representative confocal microscopy images of GFP-NLRP3 Golgi recruitment in cells transfected with wild type HA-ZDHHC3 or catalytically inactive HA-ZDHHC3[C157S]. Scale bar = 10 μm. (**F**) Quantification of GFP-NLRP3 Golgi signal in HeLaM cells co-transfected with HA-ZDHHC3 and catalytically inactive HA-ZDHHC3[C157S]. N=3. (**G**) Quantification of GFP-NLRP3 or GFP-NLRP3[HF] signal on the Golgi in HeLaM cells transiently transfected with 0.25 μg HA-ZDHHC3. N=3.

The online version of this article includes the following source data and figure supplement(s) for figure 2:

**Figure supplement 1.** The NLRP3 PB region provides an additional affinity for charged membranes.

**Figure supplement 1—source data 1.** Raw uncropped western blots for *Figure 2—figure supplement 1H*.

et al., 2017; Jennings and Linder, 2012; Ohno et al., 2006). To probe whether the Golgi localisation of NLRP3 is sensitive to ZDHHC overexpression, indicative of NLRP3 membrane binding being controlled by S-acylation, we co-expressed a library of HA-tagged ZDHHC enzymes alongside GFP-NLRP3. Overexpression of the related enzymes ZDHHC3 and ZDHHC7 both markedly enhanced recruitment of NLRP3 to the Golgi at the expense of cytosolic NLRP3 signal, with ZDHHC3 having a slightly stronger effect than ZDHHC7 (*Figure 2A and B*). To a much weaker extent, ZDHHC2, ZDHHC8 and ZDHHC9 also enhanced NLRP3 Golgi recruitment (*Figure 2A*). Unexpectedly, overexpression of the plasma membrane localised PAT ZDHHC5 decreased the amount of NLRP3 at the Golgi and instead caused NLRP3 to re-localise to the plasma membrane (*Figure 2A and B*).

In a secondary screen using lower amounts of each ZDHHC enzyme, ZDHHC3, ZDHHC5, and ZDHHC7 all altered NLRP3 localisation to a similar degree seen when using higher amounts of DNA; however, ZDHHC2, ZDHHC8 and ZDHHC9 had minimal impact on Golgi recruitment when expressed at lower levels (*Figure 2C and D*). Consistent with the results seen for GFP-tagged NLRP3, ZDHHC3 and ZDHHC5 also altered the localisation of untagged NLRP3 (*Figure 1—figure supplement 5B*). The effect of ZDHHC overexpression on NLRP3 localisation was dependent on the catalytic activity of the enzyme as catalytically inactive ZDHHC3$^{C157S}$ had no effect on recruitment on NLRP3 to the Golgi (*Figure 2E and F*). Thus, the localisation of NLRP3 can be altered by manipulating the levels and activity of ZDHHC enzymes, consistent with S-acylation controlling the sub-cellular localisation of NLRP3. Cys-130 and hydrophobic residues within helix$^{115-125}$ were both essential for the enhanced effect of ZDHHC3 and ZDHHC7 on NLRP3 Golgi recruitment and the effect of ZDHHC5 on recruitment of NLRP3 to the plasma membrane as mutating either of these features prevented NLRP3 re-localisation in response to ZDHHC overexpression (*Figure 2C and G*, *Figure 1—figure supplement 5B, C*, *Figure 1—figure supplement 6D*).

To gain further insight into how helix$^{115-125}$ and Cys-130 act together with the PB region to drive interaction of NLRP3 with intracellular membranes, we co-expressed NLRP3 containing mutations in the polybasic region with ZDHHC3 or ZDHHC5. PB mutations prevented localisation of NLRP3 to the Golgi at steady state and in response to nigericin, consistent with previous reports (*Figure 2—figure supplement 1A*; *Chen and Chen, 2018*). In agreement with the polybasic region driving affinity for highly charged membranes such as the plasma membrane and TGN (*Eisenberg et al., 2021*), sensitivity to PM localised ZDHHC5 required an intact PB region (*Figure 2—figure supplement 1B*); however, cis-Golgi localised ZDHHC3 (*Ernst et al., 2018*) was still able to recruit NLRP3 when the PB region was mutated (*Figure 2—figure supplement 1C, D*). These results suggest that helix$^{115-125}$ and Cys-130 can drive low levels of interaction with intracellular membranes independent of the polybasic region, likely through the hydrophobic residues within helix$^{115-125}$, but not at sufficient levels to show noticeable enrichment. The presence of the PB region alongside helix$^{115-125}$ boosts membrane affinity, preferentially favouring membranes with a high net negative charge, as illustrated by PB-dependent plasma membrane trapping in the presence of overexpressed HA-ZDHHC5.

## NLRP3 localisation to the Golgi in macrophages is dependent on Cys-130 and helix$^{115-125}$

As NLRP3 is predominantly expressed in myeloid cell types such as macrophages and monocytes, we stably transduced THP-1 cells with GFP-NLRP3, GFP-NLRP3$^{C130S}$ or GFP-NLRP3$^{HF}$ and assessed Golgi localisation in PMA treated THP-1 cells to determine whether Cys-130 and helix$^{115-125}$ are also required for localisation to the Golgi in macrophages. As previously described (*Andreeva et al., 2021*; *Chen and Chen, 2018*), endogenous NLRP3 localised to the perinuclear region where it partially overlapped with the *trans*-Golgi marker p230 (*Figure 3A*), with this perinuclear localisation matched by GFP-NLRP3 (*Figure 3B*). Consistent with the requirement for Cys-130 and the preceding hydrophobic residues for Golgi localisation in HeLaM cells, mutation of these residues removed the perinuclear pool of NLRP3 (*Figure 3B and C*), with GFP-NLRP3 also sensitive to overexpression of HA-tagged ZDHHC3 in a manner that was again dependent on Cys-130 (*Figure 3D*). To test the effects of enhanced levels of ZDHHC3 activity on inflammasome activation, we measured levels of nigericin induced cell death by propidium iodide (PI) staining of cells stably expressing HA-tagged ZDHHC3, as this had the strongest effect on NLRP3 Golgi recruitment in both HeLaM cells and THP-1 cells. Wild type but not catalytically inactive HA-ZDHHC3$^{C157S}$ enhanced Golgi recruitment of endogenous NLRP3 (*Figure 3E*) and markedly enhanced the percentage of cells positive for PI following nigericin treatment (*Figure 3F and*

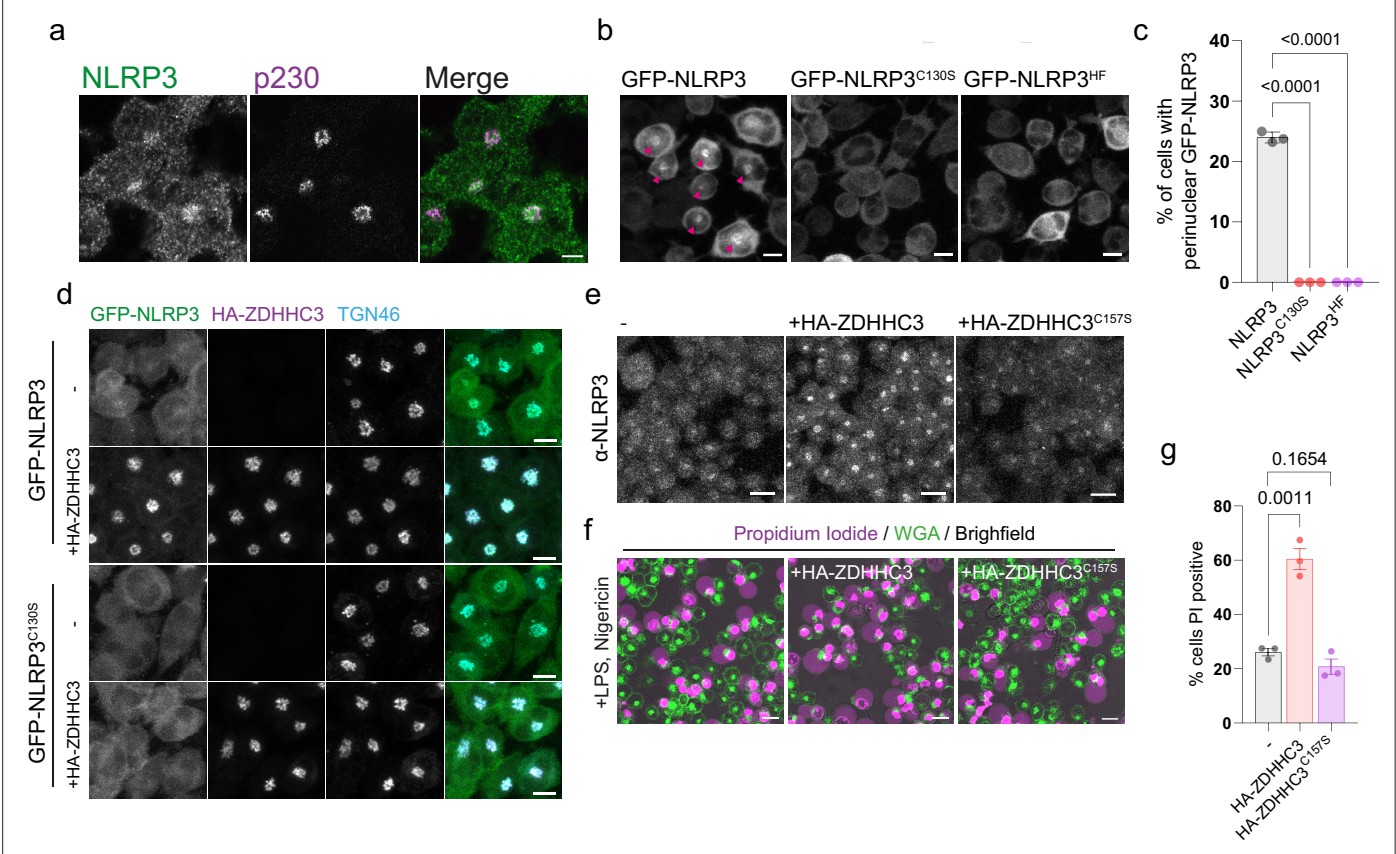

**Figure 3.** NLRP3 localisation to the Golgi is Cys-130 dependent and sensitive to ZDHHC3 levels in THP-1 macrophages. (**A**) Confocal microscopy images of LPS stimulated THP-1 cells stained with antibodies against NLRP3 and p230. Scale bar = 5 μm. (**B**) THP-1 cells stably transduced with either GFP-NLRP3, GFP-NLRP3$^{C130S}$, or GFP-NLRP3$^{HF}$ were imaged live by confocal microscopy and scored in (**C**) for the percentage of cells showing a perinuclear enrichment of GFP-NLRP3 signal. N=3. Scale bars = 10 μm. (**D**) Confocal microscopy images of stable GFP-NLRP3 or GFP-NLRP3$^{C130S}$ THP-1 cells transduced with HA-ZDHHC3 and stained for HA to visualise ZDHHC3 localisation and TGN46. Images are representative of two independent experiments. Scale bars = 10 μm. (**E**) THP-1 cells stably over-expressing HA-ZDHHC3 or HA-ZDHHC3$^{C157S}$ were stained with anti-NLRP3 antibodies to visualise NLRP3 recruitment to the peri-nuclear region. Scale bars = 20 μm. (**F**) Representative live cell confocal microscopy images of LPS primed THP-1 cells stably over-expressing HA-ZDHHC3 or HA-ZDHHC3$^{C157S}$ stimulated with 20 μM nigericin for 2 hr. Dead cells were visualised by propidium iodide staining with cell boundaries visualised through WGA-Alexa647 staining. Scale bars = 20 μm. (**G**) Quantification of the percentage of non-transduced, HA-ZDHHC3 or HA-ZDHHC3$^{C157S}$ THP-1 cells positive for propidium iodide following treatment with 20 μM nigericin for 2 hr. N=3.

**G**), demonstrating that higher levels of ZDHHC3 activity can induce a greater percentage of cells to undergo NLRP3-dependent cell death.

## NLRP3 can be S-acylated at Cys-130

To determine if the cysteine at position 130 is S-acylated, we visualised NLRP3 S-acylation using an acyl-PEG exchange (APE) assay and 5 kDa PEG maleimide, which adds an additional mass of 5 kDa to every cysteine residue that is S-acylated. Recent work has shown that NLRP3 is S-acylated within its LRR domain at position 844 by the Golgi localised PAT ZDHHC12 (*Wang et al., 2023*) and at Cys-8 within the PYD domain (*Lv et al., 2023*). Cys-844 is not involved in recruitment of NLRP3 to the Golgi (*Wang et al., 2023*), whilst Cys-8 also appears to be dispensable for this purpose (*Figure 1—figure supplement 8A*). Consistent with the presence of multiple cysteines in NLRP3 being S-acylated, label-ling of S-acylated cysteines in NLRP3 revealed two bands indicative of singly and doubly S-acylated NLRP3 species in both HeLaM and THP1 cells (*Figure 4A and C*). A large amount of NLRP3 remained resistant to hydroxylamine, suggesting a significant proportion of NLRP3 is not S-acylated at steady state. Surprisingly, mutation of Cys-130 led to only subtle changes in the S-acylation profile of NLRP3 with no loss of doubly S-acylated NLRP3 (*Figure 4A and C*), suggesting this species may represent a mix of S-acylation events at dual cysteines (i.e. Cys8/Cys844, Cys8/Cys130, Cys130/Cys844). The

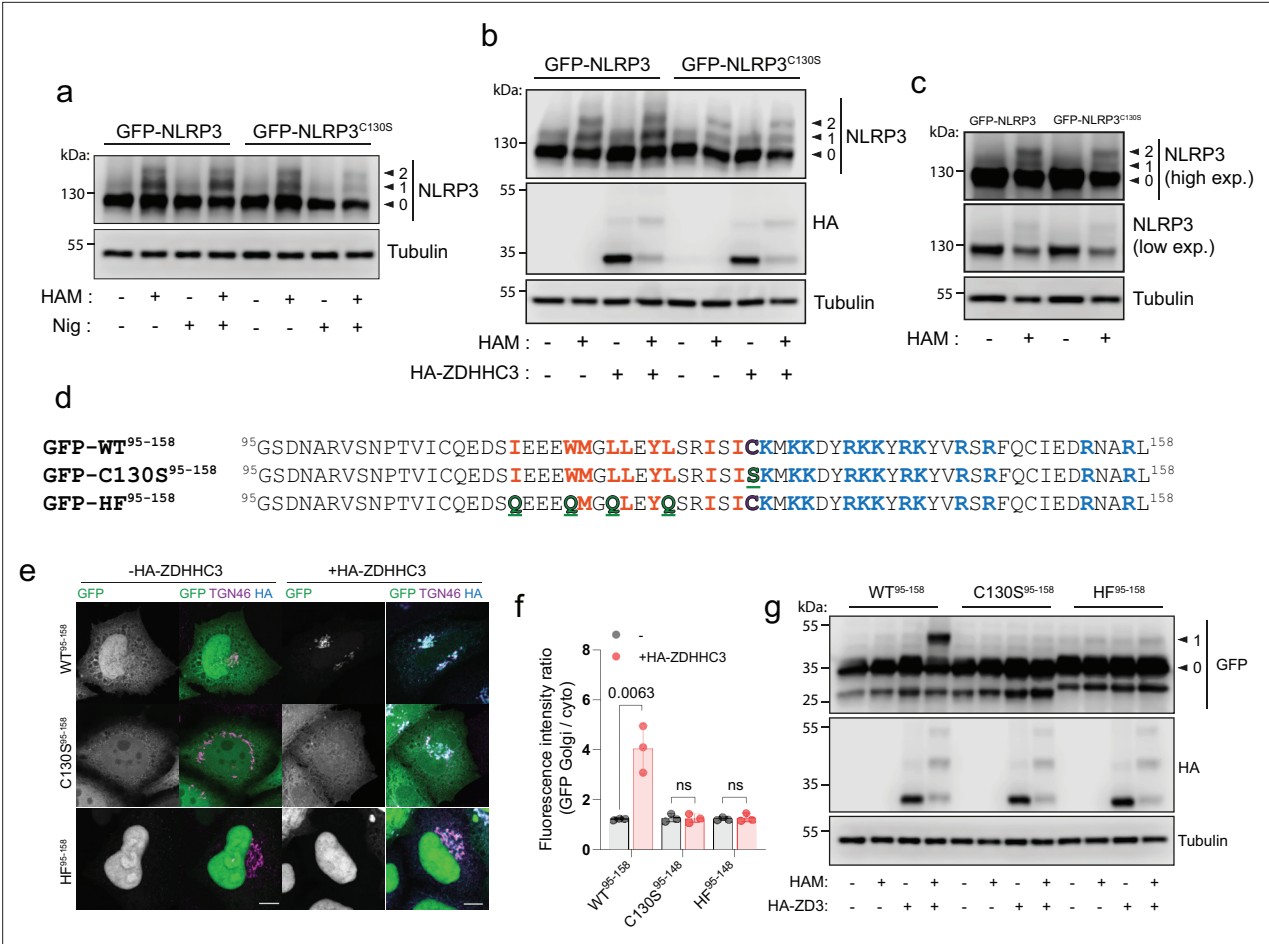

**Figure 4.** NLRP3 can be S-acylated at Cys-130. (**A**) Visualisation of S-acylation sites in HeLaM cells transiently expressing GFP-NLRP3 or GFP-NLRP3$^{C130S}$, labelled by an acyl biotin exchange assay (ABE) with 5 kDa PEG maleimide. Arrowheads indicate non-lipidated NLRP3 (0), singly S-acylated NLRP3 (1) and doubly acylated NLRP3 (2) species. HAM = hydroxylamine. (**B**) Visualisation of acylation sites by 5 kDa PEG maleimide ABE in HeLaM cells transiently expressing GFP-NLRP3 or GFP-NLRP3$^{C130S}$ and 0.25 µg HA-ZDHHC3. (**C**) Visualisation of acylation sites by 5 kDa PEG maleimide ABE in THP-1 cells stably transduced with GFP-NLRP3 or GFP-NLRP3$^{C130S}$. Blots are representative of 2 independent experiments. (**D**) Schematic of minimal GFP tagged NLRP3 sequences used and expressed in HeLaM cells to directly assess S-acylation at Cys-130. Mutations introduced are highlighted in green. (**E**) Representative confocal microscopy images of GFP-NLRP3$^{95-158}$, GFP-NLRP3$^{C130S,95-158}$ or GFP-NLRP3$^{HF,95-158}$ expressed alone or with 0.25 µg HA-ZDHHC3. N=3. Scale bars = 10 µm. (**F**) Quantification of GFP-NLRP3$^{95-158}$, GFP-NLRP3$^{C130S,95-158}$ or GFP-NLRP3$^{HF,95-158}$ signal associated with the Golgi with or without HA-ZDHHC3. (**G**) Visualisation of S-acylation sites in GFP-NLRP3$^{95-158}$ peptides expressed in HeLaM cells by ABE with or without 0.25 µg HA-ZDHHC3.

The online version of this article includes the following source data for figure 4:

**Source data 1.** Raw uncropped western blots for *Figure 4A–C, G*.

absence of a third band (Cys8, Cys130, Cys844) also suggests that S-acylation events at Cys-8, Cys-130 and Cys-844 may be mutually exclusive. Enhanced NLRP3 Golgi binding driven by ZDHHC3 overexpression and nigericin treatment both correlated with an increased amount of a singly S-acylated wild type NLRP3 species which did not occur when Cys-130 was mutated (*Figure 4A and B*).

As efforts to determine whether NLRP3 is S-acylated at Cys-130 were complicated by the presence of additional S-acylation sites in NLRP3, we next attempted to confirm S-acylation at Cys-130 using GFP-NLRP3$^{95-158}$, which incorporates helix$^{115-125}$, Cys-130 and the PB region (*Figure 4D*). GFP-NLRP3$^{95-158}$ displayed no mass shift following labelling of exposed S-acylated cysteines and did not localise to the Golgi (*Figure 4E–G*), indicating that in the context of the small peptide used for this experiment, Cys-130 is not significantly S-acylated at steady state. However, overexpression of HA-ZDHHC3 significantly increased enrichment of NLRP3$^{95-158}$ at the Golgi (*Figure 4E and F*) and induced a mass shift of NLRP3$^{95-158}$ following 5 kDa PEG maleimide labelling that was dependent on

Cys-130 and the preceding hydrophobic residues (*Figure 4G*). These results provide evidence that the linker region preceding the PB region is capable of undergoing S-acylation at Cys-130. Similar to their full-length counterparts (*Figure 2—figure supplement 1B, C*), NLRP3[95-158] peptides containing mutations in the PB region remained as sensitive to HA-ZDHHC3 overexpression as WT NLRP3 (*Figure 2—figure supplement 1E–H*).

## NLRP3 is concentrated at the Golgi by an S-acylation cycle

S-acylation of proteins can be reversed by the action of the APT and ABHD family of thioesterases which catalyse the removal of acyl groups from S-acylated cysteine residues (*Lin and Conibear, 2015*; *Tomatis et al., 2010*; *Won et al., 2018*). Small molecule inhibition of thioesterase activity by the Palmostatin family of compounds has been shown to abolish the TGN and PM localisation of H-Ras and N-Ras and leads to their redistribution across intracellular membranes (*Dekker et al., 2010*) due the stability of membrane binding provided by an acyl lipid anchor and carry over of acylated Ras proteins to the PM by vesicular transport (*Apolloni et al., 2000*). Treatment of cells with PalmostatinB had a similar effect on NLRP3, with loss of NLRP3 signal at the Golgi (*Figure 5A*) and re-localisation of NLRP3 to the plasma membrane, tubular recycling endosomes, and late endosomes (*Figure 5—figure supplement 1A–C*). Consistent with NLRP3 undergoing de-acylation, overexpression of the Golgi localised thioesterase APT2 (*Abrami et al., 2021*), but not mitochondrially localised APT1 (*Kathayat et al., 2018*; *Figure 5—figure supplement 2A*), significantly reduced localisation of NLRP3 to the Golgi in unstimulated cells (*Figure 5B*). Overexpressed APT2 had no impact on Golgi morphology, indicating that loss of NLRP3 from Golgi membranes was not due to an effect of excess APT2 on Golgi integrity (*Figure 5—figure supplement 2H–J*). Both PalmostatinB treatment and APT2 overexpression also limited the accumulation of NLRP3 on the Golgi in response to nigericin (*Figure 5C and D*), although likely for different reasons; NLRP3 remains membrane bound following PalmostatinB treatment but is physically segregated from the Golgi on post-Golgi compartments, whilst APT2 overexpression likely acts to block significant accumulation of NLRP3 on the Golgi through higher rates of NLRP3 de-acylation. The effect of APT2 overexpression appeared to be acute, suggestive of a direct action on NLRP3, as mitochondrial re-routing of APT2 from the Golgi prior to nigericin stimulation restored the enhanced recruitment of NLRP3 to the Golgi observed in response to nigericin (*Figure 5—figure supplement 2B–G*). Thus, NLRP3 dynamically associates with intracellular membranes through an S-acylation cycle.

## Nigericin treatment immobilises a fraction of NLRP3 on Golgi membranes

The amount of NLRP3 recruited to the Golgi increases following NLRP3 activation (*Figure 1—figure supplement 3A–C*; *Chen and Chen, 2018*), with this process proposed to represent a response to a danger signal common to multiple NLRP3 stimuli. The basis for enhanced Golgi recruitment is unknown. To test whether the membrane binding cycle of NLRP3 is altered in response to nigericin, we monitored the dynamics of NLRP3 membrane association by performing fluorescence recovery after photobleaching (FRAP) experiments on GFP-NLRP3 in the presence or absence of nigericin. The kinetics of NLRP3 membrane binding in untreated cells were similar to those described for mono-S-acylated N-Ras (*Rocks et al., 2005*) with a half time to recovery of 58.4 s (±9.3 ss S.D.), in agreement with NLRP3 rapidly cycling on and off the Golgi in an S-acylation cycle (*Figure 5E and F*). Nigericin treatment suppressed the fluorescence recovery; whilst GFP-NLRP3 signal on average recovered to 75% of the pre-bleach signal in untreated cells, this was reduced to 44% after nigericin treatment, indicating that a greater fraction of NLRP3 that makes up the signal in the Golgi region is immobile following nigericin treatment (*Figure 5G*). Reduced mobility of Golgi localised NLRP3 by FRAP was consistent with a mitochondrial re-routing assay using FKBP tagged GFP-NLRP3 where a proportion of perinuclear NLRP3 in nigericin treated cells remained resistant to rapamycin induced mitochondrial re-routing (*Figure 5H, I*). Changes in the membrane binding cycle of NLRP3 therefore accompany its activation, with less NLRP3 cycling off Golgi membranes following nigericin treatment.

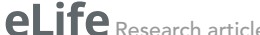

**Figure 5.** NLRP3 dynamically associates with the Golgi apparatus. (**A**) Representative confocal microscopy images of GFP-NLRP3 relative to TGN46 in HeLaM cells treated for 4 hr with 50 µM PalmostatinB or 0.4% DMSO vehicle control and quantification of GFP-NLRP3 signal associated with the Golgi with or without PalmostatinB treatment. (**B**) Representative images of GFP-NLRP3 in HeLaM cells co-expressed with 0.25 µg APT1-FLAG or 0.25 µg APT2-FLAG with quantification of GFP-NLRP3 signal associated with the Golgi in cells overexpressing APT1-FLAG or APT2-FLAG. Golgi associated GFP-NLRP3 signal is lost in cells expressing APT2-FLAG. (**C**) Representative confocal images of the effect of PalmostatinB treatment on enhanced recruitment of NLRP3 to the Golgi after nigericin treatment and quantification of perinuclear GFP-NLRP3 signal. Cells were pre-treated for 4 hr with 50 µM PalmostatinB or 0.4% DMSO control before a 1 hr stimulation with 10 µM nigericin. (**D**) Representative microscopy images of GFP-NLRP3 and APT1 or APT2-FLAG expressing cells following a 1 hr treatment with 10 µM nigericin and quantification of GFP-NLRP3 signal on the Golgi under the same conditions. (**E**) Representative images of GFP-NLRP3 expressing HeLaM cells before and after photobleaching of Golgi associated NLRP3. Scale bars in (A–E) = 10 µm. (**F**) FRAP recovery curves of GFP-NLRP3 treated with vehicle control or 10 µM nigericin for 1 hr. (**G**) Quantification of the mobile and immobile pool of GFP-NLRP3 signal in cells treated with vehicle control or 10 µM nigericin. N = 3, untreated = total of 32 cells analysed, nigericin = 29 cells. Minimum of 9 cells analysed per experiment. (**H**) Representative confocal microscopy images of FKBP tagged GFP-NLRP3 transiently expressed in HeLaM cells. Cells were treated with either vehicle control or 10 µM nigericin for 1 hr before a 15 min co-incubation with 500 nM rapamycin prior to fixation. Scale bars in main figure = 10 µm, insets = 5 µm. (**I**) Quantification of the amount of GFP-NLRP3 signal associated with the mitochondria following rapamycin treatment in the presence of absence of nigericin. Values are expressed as a percentage of the mitochondrial to cytosolic GFP-NLRP3 ratio seen in vehicle-treated control cells. N = 3.

The online version of this article includes the following figure supplement(s) for figure 5:

**Figure supplement 1.** PalmostatinB treatment relocalises NLRP3 to the plasma membrane, tubular recycling endosomes and late endosomes.

**Figure supplement 2.** The effect of APT2 on NLRP3 Golgi enrichment can be reversed.

## The thioesterase APT2 becomes segregated from NLRP3 following nigericin treatment

We next sought to understand how NLRP3 becomes immobilised on the Golgi. Golgi binding in nigericin treated cells is dependent on S-acylation at Cys-130 (*Figure 1F–I*), suggesting that increased Golgi recruitment of NLRP3 could potentially be a result of changes in the Cys-130 S-acylation cycle,

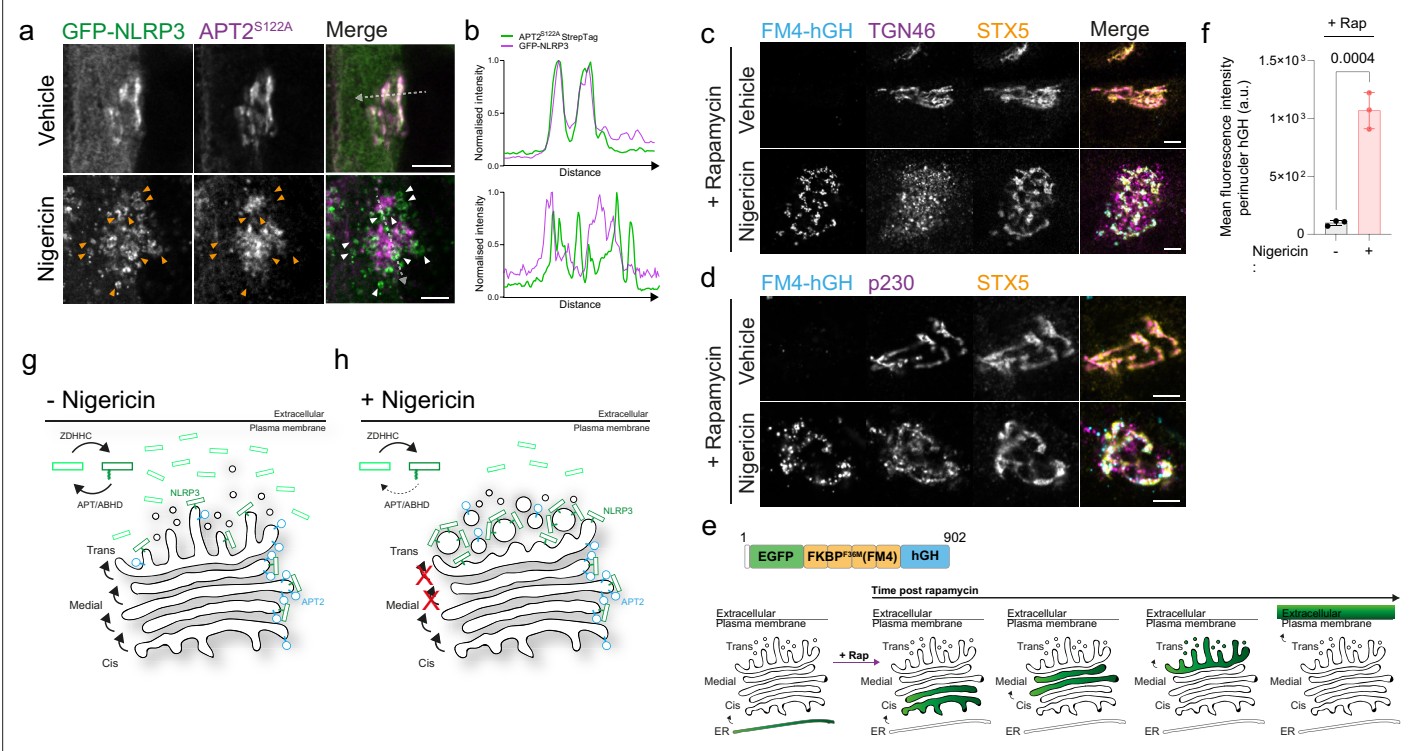

**Figure 6.** Nigericin inhibits trafficking through the Golgi apparatus and limits contact of thioesterases with NLRP3. (**A**) Representative confocal microscopy images of GFP-NLRP3 and APT2$^{S122A}$-StrepTag following a 1 hr treatment with 10 µM nigericin or vehicle control. Arrowheads point to instances of GFP-NLRP3 signal where APT2$^{S122A}$ is largely absent. Scale bars = 5 µm. (**B**) Linescan analysis of GFP-NLRP3 signal overlap with APT2$^{S122A}$ in control and nigericin-treated cells. Representative confocal microscopy images of EGFP-FM4-hGH relative to the *cis*-Golgi marker STX5 and (**C**) the TGN marker TGN46 or (**D**) the *trans*-Golgi marker p230 in HeLa cells treated with 500 nM rapamycin, in the presence or absence of 10 µM nigericin, for 1 hr. Scale bars = 5 µm. (**E**) Schematic of EGFP-FM4-hGH secretory reporter system showing the domain architecture of the EGFP-FM4-hGH reporter and position of the reporter within the secretory pathway over time after the addition of rapamycin. (**F**) Quantification of EGFP-FM4-hGH signal in the perinuclear area in cells treated with or without nigericin for 1 hr. N=3. (**G**) Proposed model for S-acylation-dependent accumulation of NLRP3 on the Golgi. In the absence of nigericin, NLRP3 shows enrichment on the Golgi due to the balance of acylation and de-acylation provided by the opposing actions of ZDHHC and APT/ABHD enzymes. (**H**) Nigericin alters the organisation of the Golgi complex and inhibits the exchange of material between Golgi cisternae. This potentially limits the amount of APT2 able to access NLRP3 leading to its gradual accumulation, likely on trans cisternae and the TGN.

The online version of this article includes the following figure supplement(s) for figure 6:

**Figure supplement 1.** Monensin enhances NLRP3 Golgi recruitment and blocks trafficking through the Golgi.

**Figure supplement 2.** Model of S-acylation dependent recruitment of NLRP3 to intracellular membranes.

**Figure supplement 3.** Palmitoyltransferase and thioesterase transcript levels in THP-1 cells and HeLa cells.

either through higher rates of S-acylation or lower rates of de-acylation. As a fraction of NLRP3 was resistant to mitochondrial rerouting (*Figure 5H, I*), suggestive of less NLRP3 being de-acylated, we tested whether immobilisation of NLRP3 may be the result of changes in the access of thioesterase enzymes to NLRP3 post nigericin treatment. The thioesterase enzyme APT2 is known to reside at the Golgi and is tethered to its cytoplasmic face via S-acylation by the *cis*-Golgi localised PATs ZDHHC3 and ZDHHC7 (*Abrami et al., 2021*; *Anwar and van der Goot, 2023*; *Ernst et al., 2018*; *Lin and Conibear, 2015*; *Vartak et al., 2014*). To examine overlap between APT2 and NLRP3, we used the catalytically inactive APT2 mutant APT2$^{S122A}$ (*Toyoda et al., 1999*; *Wang et al., 1997*) to prevent overexpressed APT2 from clearing NLRP3 from the Golgi. In untreated cells, APT2$^{S122A}$ and NLRP3 signal were closely opposed (*Figure 6A and B*). However, whilst some NLRP3 signal overlapped with APT2$^{S122A}$ after nigericin treatment, a large proportion of NLRP3 accumulated on structures that were largely devoid of APT2$^{S122A}$ (*Figure 6A and B*) indicating that APT2 becomes partially segregated from NLRP3 following nigericin treatment.

Segregation between NLRP3 and APT2 after nigericin treatment may be due to alterations in Golgi organisation and function, which could prevent mixing between Golgi compartments and encounter of NLRP3 with APT2. To investigate whether the organisation of the Golgi is affected by nigericin, we examined cis / medial, trans and TGN Golgi markers relative to one another by fluorescence microscopy. In untreated cells the cis-Golgi, trans-Golgi and TGN were maintained in close proximity as part of a well-defined Golgi ribbon (*Figure 6C, D*). In contrast, nigericin treatment induced dissociation of TGN membranes (marked by TGN46) from the cis- and medial-Golgi (*Figure 6C*), as previously described (*Chen and Chen, 2018*), with a less pronounced dissociation of p230 positive trans-Golgi membranes from the cis / medial Golgi also observed (*Figure 6D*). Nigericin treatment therefore leads to significant alterations in the organisation of distal Golgi cisternae relative to proximal cisternae.

We next investigated whether nigericin treatment also impairs Golgi function, as has previously been reported for the related ionophore monensin (*Griffiths et al., 1983*; *Tartakoff and Vassalli, 1977*). For this, we used a secretory reporter system that utilises a modified version of human growth hormone (EGFP-FM4-hGH, hereafter referred to as FM4-hGH), a luminal cargo which is normally secreted from cells via the Golgi. In this system, hGH is tagged to four repeats of FKBP containing an *F*>M mutation at position 36 (*Rollins et al., 2000*). This mutation causes FKBP to aggregate and leads to retention of FKBP-tagged hGH in the lumen of the ER (*Gordon et al., 2010*; *Rivera et al., 2000*). FM4-hGH aggregates can be solubilised by the addition of rapamycin, which allows FM4-hGH to then be exported from the ER and transit through the Golgi stack before being secreted (*Figure 6E*). Nigericin treatment in the presence of rapamycin had no effect on exit of FM4-hGH from the ER, but inhibited its trafficking through the Golgi, with FM4-hGH accumulating in structures which overlapped with the cis/ medial Golgi marker STX5 (*Figure 6C, D and F*), demonstrating that Golgi secretion and the flow of material beyond cis- and medial-Golgi cisternae is impaired by nigericin. Consistent with segregation of NLRP3 from earlier Golgi compartments, minimal overlap of STX5 and hGH with NLRP3 could be seen following nigericin treatment (*Figure 6—figure supplement 1A–D*). Thus, disruptions to Golgi organisation and function caused by nigericin may limit the ability of APT2 to come into contact with NLRP3 and de-acylate it (*Figure 6G and H*). Enhanced Golgi recruitment of NLRP3 appears to be insufficient for NLRP3 activation however as treatment of cells with the Na[+] selective ionophore monensin, which does not activate NLRP3 (*Muñoz-Planillo al., 2013*; *Perregaux et al., 1992*), also caused NLRP3 to accumulate on the Golgi (*Figure 6—figure supplement 1E, F and H*) and impaired Golgi function (*Figure 6—figure supplement 1G*).

## Discussion

In this paper, we describe the identification of additional features in NLRP3 that are necessary for localisation to the Golgi apparatus. NLRP3 Golgi binding has to date been shown to require interactions between the polybasic region of NLRP3 and PI4P (*Chen and Chen, 2018*). The NLRP3 polybasic region alone, however, is insufficient to achieve localisation of NLRP3 to the Golgi, which is consistent with studies on the mechanistic basis for stable association of peripheral membrane proteins lacking folded lipid binding domains with intracellular membranes (*Choy et al., 1999*; *Hancock et al., 1991*; *Hancock et al., 1990*). Similar to other peripheral membrane proteins that are S-acylated (*Chumpen Ramirez et al., 2020*; *Greaves et al., 2008*), electrostatic and hydrophobic interactions with intracellular membranes provided by the polybasic region and helix[115-125] likely drive an initial low-affinity membrane interaction, with stable anchoring to and enrichment on Golgi membranes provided by S-acylation at Cys-130 (*Figure 6—figure supplement 2*). Collectively, Cys-130, helix[115-125] and the polybasic region form a highly conserved functional unit that are essential for localisation of NLRP3 to the Golgi apparatus and potentially other intracellular membranes.

Despite the identification of additional structural features in NLRP3 that are essential for membrane binding, these regions alone are insufficient to achieve Golgi enrichment comparable to full length NLRP3. We could determine a minimal NLRP3 membrane binding region between residues 95–699, which incorporates all the NACHT domain, and part of the trLRR preceding the LRR domain, although this truncated protein failed to accumulate on the Golgi at steady state and after nigericin treatment to the same degree as the full-length protein. Thus, the LRR appears to play some role in the localisation of NLRP3 to intracellular membranes. Mutations in key residues involved in interactions between the LRR domains of individual NLRP3 monomers appear to limit the amount of NLRP3 on the Golgi (*Andreeva et al., 2021*). LRR-dependent formation of an oligomeric NLRP3 complex may therefore

increase affinity for intracellular membranes through an avidity-based effect dependent on multiple monomers working together within a larger NLRP3 oligomer. The Cryo-EM structures of NLRP3 are interesting in this respect, as the polybasic regions from multiple monomers appear to be organised within each pentamer / hexamer to form an expanded positively charged docking site around Cys-130 and helix[115-125] (*Andreeva et al., 2021*; *Hochheiser et al., 2022*; *Figure 1—figure supplement 1D–G*). Membrane contact driven through polybasic / helix[115-125] based multi-valent interactions could then be stabilised by S-acylation at Cys-130 at one or multiple monomers.

Consistent with S-acylation controlling NLRP3 localisation, we provide evidence that NLRP3 is highly sensitive to levels of ZDHHC3, ZDHHC5, ZDHHC7 and APT2, and that overexpression of ZDHHC3 can sensitise macrophages to nigericin. However, we did not identify the endogenous enzymes responsible for stable NLRP3 recruitment to the Golgi through S-acylation and de-acylation, although both ZDHHC3/5/7 and APT2 transcripts are expressed at high levels in THP-1 cells relative to other ZDHHC and APT/ABHD family members (*Figure 6—figure supplement 3*). In agreement with our results showing that localisation of NLRP3 to the Golgi can be controlled by the levels of ZDHHC3/7, a recent pre-print demonstrated that knockout of ZDHHC7 prevents localisation of NLRP3 to the Golgi and limits NLRP3 activation, suggesting that ZDHHC7 is responsible for NLRP3 S-acylation at Cys-130 in macrophages (*Yu et al., 2023*).

Our data showing sensitivity of NLRP3 to ZDHHC enzymes localised to distinct organelles (Golgi localised ZDHHC3/7 vs plasma membrane localised ZDHHC5) is intriguing as it suggests NLRP3 can be recruited to different sub-cellular locations through S-acylation at Cys-130, dependent upon PAT abundance and the affinity of NLRP3 for different intracellular membranes dictated by the polybasic region and helix[115-125]. This may help to reconcile our own observations demonstrating NLRP3 association with the Golgi through an S-acylation cycle controlled by Golgi localised palmitoylation machinery, with recent work describing NLRP3 localisation to early and late endosomes in response to NLRP3 agonists, where it is proposed to sense disruptions in endocytic trafficking and ER-endosome contact sites (*Lee et al., 2023*; *Zhang et al., 2023*). Alterations in the levels of charged lipids on endosomes caused by NLRP3 stimuli, as has previously been described (*Lee et al., 2023*; *Zhang et al., 2023*), could potentially allow NLRP3 to visit endosomal membranes and be trapped there by an endosomally localised PAT. Re-localisation of NLRP3 upon overexpression of ZDHHC5 supports the idea that a switch in the use of differentially localised PATs could re-route NLRP3 to different sub-cellular compartments including early endosomes, where ZDHHC5 has been proposed to function (*Breusegem and Seaman, 2014*). An alternative route of endosomal recruitment could also be through a broader inhibition of thioesterase activity, as we observed NLRP3 co-localisation with late endosomes and Rab8[+] tubular recycling endosomes when de-acylation was inhibited by PalmostatinB, although this re-localisation occurred over the timescale of a number of hours.

Membrane binding is required for formation of an inactive oligomeric NLRP3 complex (*Andreeva et al., 2021*). Disruption of oligomer formation, either through mutation of residues involved in contacts between individual monomers or mutation of the NLRP3 polybasic region, limits NLRP3 activation, suggesting that membrane dependent oligomer formation is essential for sensing of NLRP3 stimuli (*Andreeva et al., 2021*; *Chen and Chen, 2018*). Regulation of access to the Golgi through S-acylation at Cys-130 is likely to be important for oligomer formation and may represent another aspect of regulatory control over NLRP3 activation. The steady state balance of S-acylation and de-acylation at Cys-130 may keep levels of Golgi localised NLRP3 below a critical threshold required for further NLRP3 oligomerisation or access to interaction partners and post translational modifications needed for optimal activation. The importance of membrane binding for NLRP3 inflammasome formation in humans remains unclear. In mice, deletion of residues 92–120, which removes part of helix[115-125], impairs IL1β release, with longer deletions (Δ92–132 and Δ92–148) completely removing Cys-130 and helix[115-125] producing an even stronger reduction in inflammasome activation (*Tapia-Abellán et al., 2021*). In contrast, deletion of exon-3 (spanning residues 95–134) from human NLRP3 delays but does not completely block NLRP3 activation in human macrophages (*Mateo-Tórtola et al., 2023*). In light of the latter study, S-acylation at Cys-130 and binding to the Golgi may accelerate the process of inflammasome formation or increase sensitivity to NLRP3 agonists, although activation of human NLRP3 can still seemingly occur in the absence of Golgi binding.

Based on our mechanism proposing that nigericin induced alterations in Golgi organisation and trafficking may limit access of thioesterases to NLRP3, enhanced binding of NLRP3 to the Golgi

would likely occur in response to treatments that perturb Golgi function in a similar manner. However, enhanced recruitment of NLRP3 to the Golgi is not necessarily indicative of NLRP3 activation, given that monensin also triggers accumulation of NLRP3 on the Golgi, although it is worth nothing that monensin can potentiate the effect of weaker NLRP3 agonists (*Lee et al., 2023*). Whilst the exact reason NLRP3 has evolved to bind to intracellular membranes at present remains unknown, our results nonetheless provide new insight into the mechanisms underpinning membrane recruitment of NLRP3.

## Methods

### Plasmids

pCMV-APT1-Myc-FLAG and pCMV-APT2-Myc-FLAG were gifts from Professor Gisou Van der Goot (École polytechnique fédérale de Lausanne (EPFL), Switzerland). pEGFP-+8 pre was a gift from Professor Sergio Grinstein (Hospital for Sick Kids, Ontario, Canada). The HA tagged ZDHHC library containing ZDHHC1-23 was a gift from Professor Masaki Fukata (Division of Membrane Physiology, National Institute for Physiological Sciences, Japan). mCherry-P4M-SidM was purchased from Addgene (Plasmid #51471). pEGFP-NRas was a gift from Professor Ian Prior (University of Liverpool, UK). pMito-mCherry-FRB was a gift from Professor Stephen Royle, (University of Warwick, UK). All other plasmids used in this study were cloned using standard molecular biology techniques. In brief, GFP tagged NLRP3 and untagged NLRP3 were amplified and cloned into pIRES Neo between BamHI and NotI sites. For FKBP tagging of GFP tagged WT NLRP3, FKBP was cloned into AgeI and BamHI sites upstream of GFP-NLRP3 in pIRES Neo. NLRP3$^{HF}$, NLRP3$^{PB-1}$ and NLRP3$^{PB-2}$ were made by extension overlap PCR from wild type GFP tagged NLRP3 and cloned into pIRES Neo between BamHI and NotI sites. For point mutations in NLRP3 and ZDHHC3, an NEB Q5 SDM kit (NEB, E0554S) was used, followed by sequencing of clones to confirm the presence of the correct mutation. For generation of APT2-StrepTag-FKBP, StrepTag-FKBP was cloned between PmeI and NotI sites in at the C-terminus of APT2 in pCMV. For generation of the minimal NLRP3 constructs spanning residues 95–158 and variants thereof, GFP was first cloned into pIRES neo between AgeI and BamHI sites before insertion of the NLRP3$^{95-158}$ sequences between BamHI and NotI sites. For pLXIN constructs, all inserts were cloned between BamHI and NotI sites in pLXIN. Primer details for all constructs described are available upon request.

### Cell culture

HeLaM cells were originally obtained from the laboratory of Margaret S. Robinson (Cambridge Institute of Medical Research, University of Cambridge, UK). HEK-293T cells were obtained from ATCC. THP-1 cells were a kind gift from Professor Paul Lehner, University of Cambridge. HeLa cells stably expressing EGFP-FM4-hGH were generated in house and described previously (*Gordon et al., 2010*; *Gordon et al., 2021*). HeLa, HeLaM and HEK-293T cells were grown in Dulbeccos modified Eagles medium (DMEM) (Merck, D6429-500ML) supplemented with 10% Foetal bovine serum (FBS) (Merck, F4135-500ML), 100 IU/ml pencillin and 100 µg / ml streptomycin (Merck, G1146-100ML) at 37 °C in a 5% CO$_2$ humidified incubator. THP-1 cells were grown in Roswell Park Memorial Institute (RPMI) 1640 medium (Merck, R8758-500ML) supplemented with 10% FBS, 100 IU / ml pencillin, 100 µg / ml streptomycin and 2 mM L-glutamine (Fisher Scientific Ltd, 10691233) at 37 °C in a 5% CO$_2$ humidified incubator. Additional reagents used as part of cell culture experiments: monensin (Merck, M5273), brefeldin A (Biolegend, 420601), rapamycin (Apex Bio, A8167).

### Generation of stable cell lines

For generation of all stable THP-1 cell lines, a second-generation retroviral vector system was used. Briefly, HEK-293T cells seeded into 12 well plates were transfected with the retroviral packaging plasmid pUMVC (Addgene: 8449), pLXIN (Clontech, 631501) containing the gene of interest and pCMV VSV-G at a ratio of 5:5:1 with polyethylenimine (PEI) (1 µg DNA to 3 µg PEI). Forty-eight hr post transfection, cell culture media was harvested and clarified by centrifugation to pellet loose cells. The supernatant was collected and, after addition of polybrene to a final concentration of 10 µg / ml, viral media was then added to THP-1 cells. The THP-1 cell-viral media suspension was then centrifuged at 2500 x *g* for 90 min at room temperature. Cells were incubated at 37 °C and successfully transduced cells selected by addition of 1 mg / ml G418 (SLS, 4727878001) to the cell media for 1 week.

## Transfections and mmunofluorescence

HeLaM cells were seeded onto glass coverslips in 12-well plates at a density of $2.25 \times 10^5$ cells / well and left to adhere overnight. The next day, cells were transfected with plasmid DNA using Viafect (E4982, Promega) at a ratio of 3:1 viafect to DNA, according to the manufacturer's instructions. Cells were then incubated overnight at 37 °C and processed the next day. For HeLaM and THP-1 cell staining's, cells were pre-fixed by addition of an equivalent volume of 4% paraformaldehyde (PFA, Park Scientific Limited, 04018–4) to cell culture media for 1 mine. PFA-media solution was aspirated, and cells were then incubated with 4% PFA for 10 min at room temperature. After aspirating, cells were incubated with 100 mM Glycine (Merck, G7126) in PBS for 3 min to quench PFA. Glycine was then aspirated and cells permeabilised and blocked for 5 min with PBS containing 1% bovine serum albumin (BSA; BP1600-100, Thermo Fisher Scientific) and 0.1% saponin (84510–100 G, Merck; IF buffer). Cells were incubated with primary antibody solutions in IF buffer for 30 min at 37 °C before being washed three times with IF buffer. Cells were then incubated with secondary antibodies in IF buffer for a further 30 min at 37 °C. Following secondary antibody incubation cells were washed a further 3 times with IF buffer before a final wash in PBS. Coverslips were mounted onto glass slides using prolong gold antifade (ThermoFisher Scientific, P36930). Coverslips were imaged on a Nikon A1 upright confocal microscope with a 60 x, NA 1.4, Plan Apochromat oil immersion objective with approximately 1.4 x digital zoom added at a pixel resolution of 1024x1,024.

## Fluorescence recovery after photobleaching (FRAP)

Cells were seeded overnight at a density of $2.25 \times 10^5$ / quarter of a cellview cell culture dish (Greiner Bio, 627870) and transfected with GFP-NLRP3 the next day. FRAP experiments were performed on a Zeiss LSM880 airyscan confocal microscope using a 63X1.4 NA Plan Apochromat oil immersion objective with a temperature-controlled stage set to 37 °C. For FRAP experiments, the entire Golgi region was photobleached using the 405 nm laser-line before images were acquired at 3 s intervals for a period of 6 min. Recovery curves were generated by measurement of GFP intensity in the bleached area at each individual timepoint and were adjusted for fluorescence decay caused by prolonged imaging during the recovery period. The immobile and mobile pools of NLRP3 were calculated by the following formula:

$$\text{Immobile fraction} = 1 - (\text{MFI}^{\text{final}} - \text{MFI}^{\text{bleach}})/(\text{MFI}^{\text{pre-bleach}}\text{MFI}^{\text{bleach}})$$

$$\text{Mobile fraction} = (\text{MFI}^{\text{final}} - \text{MFI}^{\text{bleach}})/(\text{MFI}^{\text{pre-bleach}} - \text{MFI}^{\text{bleach}})$$

where $\text{MFI}^{\text{final}}$ corresponds to the mean fluorescence intensity (MFI) in the bleach area at the final recovery timepoint post-bleach, $\text{MFI}^{\text{bleach}}$ corresponds to the MFI in the bleach area immediately after photobleaching and $\text{MFI}^{\text{pre-bleach}}$ corresponds to the MFI in the bleach area prior to photobleaching. Recovery half times were calculated using EasyFRAP software (*Koulouras et al., 2018*).

## Inflammasome assays

THP1 cells were seeded into 96 well plates in RPMI containing 100 ng / μl PMA (PMA) (Merck, P1585) and left to adhere overnight at 37 °C. The next day, PMA was aspirated and cells incubated with RPMI containing 1 μg / ml LPS (Merck, L2630) for 3 hr. Prior to nigericin stimulation, cells were washed once with PBS and incubated with a solution containing LPS and 1 μg / ml wheat germ agglutinin-Alexa-647 (WGA) (Thermo Fisher Scientific, W3246,) for 10 min to visualise the plasma membrane and facilitate quantification. Cells were washed once with PBS and then incubated with RPMI containing 1 μg / ml propidium iodide (PI) (Merck, 81845) and nigericin (Merck, N7143) (20 μg / ml) for 2 hr before imaging live on a Nikon A1 upright confocal microscope. Per experiment, a minimum of four separate fields of view per cell line were acquired and the number of live cells and dead cells counted manually based on the presence of propidium iodide.

## PEG switch assay

HeLaM cells were seeded into 12 well plates and transfected as described above. The next day cells were lysed in 200 μl of PEG switch sample buffer (PS buffer) (1% SDS, 50 mM HEPES) supplemented with 100 mM NEM (Merck, E3876), 1 mM PMSF (Merck, P7626-1G) and 1 X protease inhibitor

cocktail (Merck, 11873580001) for 30 min at room temperature. Cell lysates were incubated at 40 °C with shaking for 4 hr to block free cysteine residues. Following this, to remove NEM and precipitate proteins, 4 volumes of 100% acetone were added to lysates and incubated for 1 hr at –20 °C before lysates were centrifuged at 15,000 x *g* for 10 min to pellet precipitated proteins. The acetone-sample buffer mix was aspirated, and protein pellets were washed our times with 70% acetone to remove residual NEM. After the final wash, acetone was aspirated and protein pellets dried at room temperature for 10 min. Pellets were resuspended in 200 µl PS buffer before the sample was split in two and either 0.4 M hydroxylamine (SLS, 159417–100 G) or water added. Samples were incubated for 1 hr at 37 °C before proteins were again acetone precipitated for 1 hr as described above. Pellets were washed once more with 70% acetone, air-dried for 10 min and resuspended in 100 µl PS buffer. Two mM PEG-maleimide was then added to label exposed S-acylated cysteine residues and samples incubated for 1 hr at 37 °C. Following addition of SDS-PAGE sample buffer the samples were boiled for 5 min at 95 °C and resolved by SDS-PAGE.

## Western blotting

SDS-PAGE sample buffer (BioRad, 1610747) supplemented with 5% (v/v) mercaptoethanol was added to samples lysed in PS buffer before boiling for 5 min at 95 °C. Samples were then loaded onto an appropriate percentage Tris-Glycine polyacrylamide gel and proteins resolved by SDS-PAGE. Gels were transferred to PVDF membrane using a BioRad semi-dry transfer system. Following transfer, membranes were blocked in 5% milk in PBS containing 0.1% Triton X-100 (PBSTx) for a minimum of 30 min. Blocked membranes were incubated with primary antibodies diluted in PBSTx and 5% milk overnight at 4 °C. The next day, unbound antibodies were removed by 3x5 min washes in PBSTx before incubation with secondary antibodies in PBSTx containing 5% milk for 1 hr with rotation at room temperature. Membranes were again washed three times for 5 min with PBSTx and signal detected using BioRad ECL chemiluminescence substrate and the LiCor c-digit system.

## Quantification and statistical analysis

For quantification of Golgi associated NLRP3 signal in untreated and nigericin treated cells, the Golgi region was identified using a Golgi marker which remains associated with the Golgi following nigericin treatment (either GM130 or p230) and the amount of NLRP3 associated measured through manual segmentation of the Golgi region in FIJI (*Schindelin et al., 2012*). An equivalent area of GFP-NLRP3 signal in the cytosol away from the perinuclear region was measured and the ratio between the Golgi NLRP3 signal and cytosolic NLRP3 signal calculated for each cell expressing GFP-NLRP3. All N numbers represent a minimum of 20 cells quantified per experiment with each graphed datapoint representing an independent experiment performed on separate days. All graphs included in figures were made using the GraphPad Prism software package. All statistical comparisons between groups were made using an unpaired t-test in GraphPad Prism 9.0.

## Materials availability

Newly created materials for this study are available upon request from the corresponding authors.

## Antibodies

| Antibody | Supplier / Cat # | RRID | Host species / type |
|---|---|---|---|
| TGN46 | Bio-Rad Cat# AHP500GT | RRID:AB_2203291 | Sheep polyclonal |
| HA | BioLegend Cat# 901502 | RRID:AB_2565007 | Mouse monoclonal |
| NLRP3 | Abcam Cat# ab4207 | RRID:AB_955792 | Goat polyclonal |
| NLRP3 | AdipoGen Cat# AG-20B-0014 | RRID:AB_2490202 | Mouse monoclonal |
| Rab8 | Cell Signalling Tech. Cat# 6975 | RRID:AB_10827742 | Rabbit monoclonal |

*Continued on next page*

*Continued*

| Antibody | Supplier / Cat # | RRID | Host species / type |
|---|---|---|---|
| CD63 | DSHB Cat# H5C6 | RRID:AB_528158 | Mouse monoclonal |
| p230 | BD Biosciences Cat# 611280 | RRID:AB_398808 | Mouse monoclonal |
| Tubulin | Proteintech Cat# 66240–1-Ig | RRID:AB_2881629 | Mouse monoclonal |
| STX5 | Synaptic Systems Cat# 110 053 | RRID:AB_887800 | Rabbit polyclonal |
| RCAS1 | Cell Signalling Tech Cat# 12290 | RRID:AB_2736985 | Rabbit monoclonal |
| EEA1 | BD Biosciences Cat# 610457 | RRID:AB_397830 | Mouse monoclonal |
| GM130 | BD Biosciences Cat# 610823 | RRID:AB_398142 | Mouse monoclonal |
| anti-mouse-HRP | Jackson ImmunoResearch Labs Cat# 115-035-008 | RRID:AB_2313585 | Goat polyclonal |
| anti-rabbit-HRP | Jackson ImmunoResearch Labs Cat# 111-035-144 | RRID:AB_2307391 | Goat polyclonal |
| Anti-rabbit Alexa-488 | Thermo Fisher Scientific Cat# A-21206 | RRID:AB_2535792 | Donkey polyclonal |
| Anti-rabbit Alexa-594 | Thermo Fisher Scientific Cat# A-21207 | RRID:AB_141637 | Donkey polyclonal |
| Anti-rabbit Alexa-647 | Thermo Fisher Scientific Cat# A-31573 | RRID:AB_2536183 | Donkey polyclonal |
| Anti-mouse Alexa-488 | Thermo Fisher Scientific Cat# A-21202 | RRID:AB_141607 | Donkey polyclonal |
| Anti-mouse Alexa-594 | Thermo Fisher Scientific Cat# A-21203 | RRID:AB_2535789 | Donkey polyclonal |
| Anti-mouse Alexa-647 | Thermo Fisher Scientific Cat# A-31571 | RRID:AB_162542 | Donkey polyclonal |
| Anti-rabbit Alexa-647 | Thermo Fisher Scientific Cat# A-21443 | RRID:AB_2535861 | Chicken polyclonal |
| Anti-mouse Alexa-647 | Thermo Fisher Scientific Cat# A-21463 | RRID:AB_2535869 | Chicken polyclonal |
| Anti-goat Alexa-594 | Thermo Fisher Scientific Cat# A-11058 | RRID:AB_2534105 | Donkey polyclonal |

## Plasmids

| Plasmid | Reference | Backbone |
|---|---|---|
| APT1-Myc-FLAG | Origene Cat#: RC202029 | pCMV6 |
| APT2-Myc-FLAG | Origene Cat#: RC202021 | pCMV6 |
| EGFP-+8 pre | *Yeung et al., 2008* | pEGFP |
| HA-ZDHHC1-23 | *Fukata et al., 2004* | pEF-Bos |
| mCherry-P4M-SidM | Addgene (Plasmid #51471) | pmCherry-N1 |
| EGFP-NRas | *Laude and Prior, 2008* | pEGFP |
| Untagged NLRP3 | This study | pIRES-Neo |
| Untagged NLRP3$^{C130S}$ | This study | pIRES-Neo |
| GFP-NLRP3 | This study | pIRES-Neo |
| GFP-NLRP3$^{C130S}$ | This study | pIRES-Neo |
| GFP-NLRP3$^{HF}$ | This study | pIRES-Neo |
| GFP-NLRP3$^{PB-1}$ | This study | pIRES-Neo |
| GFP-NLRP3$^{PB-2}$ | This study | pIRES-Neo |
| GFP-NLRP3$^{95-1034}$ | This study | pIRES-Neo |

*Continued on next page*

*Continued*

| Plasmid | Reference | Backbone |
|---|---|---|
| GFP-NLRP3[110-1034] | This study | pIRES-Neo |
| GFP-NLRP3[1-699] | This study | pIRES-Neo |
| GFP-NLRP3[1-680] | This study | pIRES-Neo |
| GFP-NLRP3[95-699] | This study | pIRES-Neo |
| GFP-NLRP3[95-158] | This study | pIRES-Neo |
| GFP-NLRP3[95-158-(C130S)] | This study | pIRES-Neo |
| GFP-NLRP3[95-158-(HF)] | This study | pIRES-Neo |
| GFP-NLRP3[95-158-(PB-1)] | This study | pIRES-Neo |
| GFP-NLRP3[95-158-(PB-2)] | This study | pIRES-Neo |
| GFP-NLRP3[95-158-CVIM] | This study | pIRES-Neo |
| GFP-NLRP3[95-158-CVIM-(C130S)] | This study | pIRES-Neo |
| GFP-NLRP3[95-130] | This study | pIRES-Neo |
| pEGFP-N3 | Addgene (Plasmid #2493) | pEGFP-N3 |
| GFP-NLRP3[131-158] | This study | pEGFP-N3 |
| GFP-CVIM | This study | pEGFP-N3 |
| EGFP-NLRP3[131-158-CVIM] | This study | pEGFP-N3 |
| FKBP-GFP-NLRP3 | This study | pIRES-Neo |
| APT1-StrepTag-FKBP | This study | pCMV |
| APT2-StrepTag-FKBP | This study | pCMV |
| APT2[S122A]-StrepTag-FKBP | This study | pCMV |
| HA-ZDHHC3[C157S] | This study | pEF-Bos |
| HA-ZDHHC3 | This study | pLXIN |
| HA-ZDHHC3[C157S] | This study | pLXIN |
| GFP-NLRP3 | This study | pLXIN |
| GFP-NLRP3[C130S] | This study | pLXIN |
| GFP-NLRP3[HF] | This study | pLXIN |
| mCherry-FRB | Addgene (Plasmid #59352) | pMito |

## Acknowledgements

We are grateful to the University of Sheffield Wolfson Light microscopy facility for technical support with microscopy experiments. We also thank Dr Mark Collins (University of Sheffield), for advice on acylation assays and Professor Martin Lowe (University of Manchester) and Dr Xiaoming Fang (University of Cambridge) for helpful discussions and advice on the manuscript. AP and DW are supported by the BBSRC (BB/S009566/1).

## Additional information

### Funding

| Funder | Grant reference number | Author |
| --- | --- | --- |
| Biotechnology and Biological Sciences Research Council | BB/S009566/1 | Andrew A Peden |

The funders had no role in study design, data collection and interpretation, or the decision to submit the work for publication.

### Author contributions

Daniel M Williams, Conceptualization, Formal analysis, Investigation, Writing – original draft, Writing – review and editing; Andrew A Peden, Conceptualization, Funding acquisition, Writing – review and editing

### Author ORCIDs

Daniel M Williams ⓘ https://orcid.org/0000-0001-8483-021X
Andrew A Peden ⓘ https://orcid.org/0000-0003-0144-7712

Reviewer #2 (Public Review): https://doi.org/10.7554/eLife.94302.3.sa1
Author response https://doi.org/10.7554/eLife.94302.3.sa2

## Additional files

### Supplementary files
• MDAR checklist

### Data availability
Source data containing uncropped western blots are provided.

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
