## [Editor Report · eLife assessment]

This **important** paper implicates S-acylation of Cys-130 in recruitment of the inflammasome receptor NLRP3 to the Golgi, and it provides **convincing** evidence that S-acylation plays a key role in response to the stress induced by nigericin treatment. While Cys-130 does seem to play a previously unappreciated role in membrane association of NLRP3, further work will be needed to clarify the details of the mechanism.

---

## [Referee Report · Reviewer #2 (Public Review)]

This paper examines the recruitment of the inflammasome seeding pattern recognition receptor NLRP3 to the Golgi. Previously, electrostatic interactions between the polybasic region of NLRP3 and negatively charged lipids were implicated in membrane association. The current study concludes that reversible S-acylation of the conserved Cys-130 residue, in conjunction with upstream hydrophobic residues plus the polybasic region, act together to promote Golgi localization of NLRP3, although additional parts of the protein are needed for full Golgi localization. Treatment with the bacterial ionophore nigericin inhibits membrane traffic and apparently prevents Golgi-associated thioesterases from removing the acyl chain, causing NLRP3 to become immobilized at the Golgi. This mechanism is put forth as an explanation for how NLRP3 is activated in response to nigericin.

The experiments are generally well presented. It seems likely that Cys-130 does indeed play a previously unappreciated role in Golgi association of NLRP3. However, the evidence for S-acylation at Cys-130 is largely indirect, and the process by which nigericin enhances membrane association is not yet fully understood. Therefore, this interesting study points the way for further analysis.

---

## [Author Response]

The following is the authors’ response to the original reviews.

**Public Reviews:**

**Reviewer #1 (Public Review):**
This is an interesting study investigating the mechanisms underlying membrane targeting of the NLRP3 inflammasome and reporting a key role for the palmitoylation-depalmitoylation cycle of cys130 in NRLP3. The authors identify ZDHHC3 and APT2 as the specific ZDHHC and APT/ABHD enzymes that are responsible for the s-acylation and de-acylation of NLRP3, respectively. They show that the levels of ZDHHC3 and APT2, both localized at the Golgi, control the level of palmitoylation of NLRP3. The S-acylation-mediated membrane targeting of NLRP3 cooperates with polybasic domain (PBD)-mediated PI4P-binding to target NLRP3 to the TGN under steady-state conditions and to the disassembled TGN induced by the NLRP3 activator nigericin.However, the study has several weaknesses in its current form as outlined below.(1) The novelty of the findings concerning cys130 palmitoylation in NLRP3 is unfortunately compromised by recent reports on the acylation of different cysteines in NLRP3 (PMID: 38092000), including palmitoylation of the very same cys130 in NLRP3 (Yu et al https://doi.org/10.1101/2023.11.07.566005), which was shown to be relevant for NLRP3 activation in cell and animal models. What remains novel and intriguing is the finding that NLRP3 activators induce an imbalance in the acylation-deacylation cycle by segregating NLRP3 in late Golgi/endosomes from de-acylating enzymes confined in the Golgi. The interesting hypothesis put forward by the authors is that the increased palmitoylation of cys130 would finally contribute to the activation of NLRP3. However, the authors should clarify the trafficking pathway of acylated-NLRP3. This pathway should, in principle, coincide with that of TGN46 which constitutively recycles from the TGN to the plasma membrane and is trapped in endosomes upon treatment with nigericin.

We think the data presented in our manuscript are consistent with the majority of S-acylated NLRP3 remaining on the Golgi via S-acylation in both untreated and nigericin treated cells. We have performed an experiment with BrefeldinA (BFA), a fungal metabolite that disassembles the Golgi without causing dissolution of early endosomes, that further supports the conclusion that NLRP3 predominantly resides on Golgi membranes pre and post activation. Treatment of cells with BFA prevents recruitment of NLRP3 to the Golgi in untreated cells and blocks the accumulation of NLRP3 on the structures seen in the perinuclear area after nigericin treatment (see new Supplementary Figure 4A-D). We do see some overlap of NLRP3 signal with TGN46 in the perinuclear area after nigericin treatment (see new Supplementary Figure 2E), however this likely represents TGN46 at the Golgi rather than endosomes given that the NLRP3 signal in this area is BFA sensitive. As with 2-BP and GFP-NLRP3C130S, GFP-NLRP3 spots also form in BFA / nigericin co-treated cells but not with untagged NLRP3. These spots also do not show any co-localisation with EEA1, suggesting that under these conditions, endosomes don’t appear to represent a secondary site of NLRP3 recruitment in the absence of an intact Golgi. However, we cannot completely rule out that some NLRP3 may recruited to endosomes at some point during its activation.

(2) To affect the S-acylation, the authors used 16 hrs treatment with 2-bromopalmitate (2BP). In Figure 1f, it is quite clear that NLRP3 in 2-BP treated cells completely redistributed in spots dispersed throughout the cells upon nigericin treatment. What is the Golgi like in those cells? In other words, does 2-BP alter/affect Golgi morphology? What about PI4P levels after 2-BP treatment? These are important missing pieces of data since both the localization of many proteins and the activity of one key PI4K in the Golgi (i.e. PI4KIIalpha) are regulated by palmitoylation.

We thank the reviewer for highlighting this point and agree that it is possible the observed loss of NLRP3 from the Golgi might be due to an adverse effect of 2-BP on Golgi morphology or PI4P levels. We have tested the effect of 2-BP on the Golgi markers GM130, p230 and TGN46. 2BP has marginal effects on Golgi morphology with cis, trans and TGN markers all present at similar levels to untreated control cells (Supplementary Figure 2B-D). We also tested the effect of 2-BP on PI4P levels using mCherry-P4M, a PI4P biosensor. Surprisingly, as noted by the reviewer, despite recruitment of PI4K2A being dependent on S-acylation, PI4P was still present on the Golgi after 2-BP treatment, suggesting that a reduction in Golgi PI4P levels does not underly loss of NLRP3 from the Golgi (Supplementary Figure 2A). The pool of PI4P still present on the Golgi following 2-BP treatment is likely generated by other PI4K enzymes that localise to the Golgi independently of S-acylation, such as PI4KIIIB. We have included this data in our manuscript as part of a new Supplementary Figure 2.

(3) The authors argue that the spots observed with NLRP-GFP result from non-specific effects mediated by the addition of the GFP tag to the NLRP3 protein. However, puncta are visible upon nigericin treatment, as a hallmark of endosomal activation. How do the authors reconcile these data? Along the same lines, the NLRP3-C130S mutant behaves similarly to wt NLRP3 upon 2-BP treatment (Figure 1h). Are those NLRP3-C130S puncta positive for endosomal markers? Are they still positive for TGN46? Are they positive for PI4P?

This is a fair point given the literature showing overlap of NLRP3 puncta formed in response to nigericin with endosomal markers and the similarity of the structures we see in terms of size and distribution to endosomes after 2BP + nigericin treatment. We have tested whether these puncta overlap with EEA1, TGN46 or PI4P (Supplementary Figure 2A, E-G). The vast majority of spots formed by GFP-NLRP3 co-treated with 2-BP and nigericin do not co-localise with EEA1, TGN46 or PI4P. This is consistent with these spots potentially being an artifact, although it has recently been shown that human NLRP3 unable to bind to the Golgi can still respond to nigericin (Mateo-Tórtola et al., 2023). These puncta might represent a conformational change cytosolic NLRP3 undergoes in response to stimulation, although our results suggest that this doesn’t appear to happen on endosomes.

(4) The authors expressed the minimal NLRP3 region to identify the domain required for NLRP3 Golgi localization. These experiments were performed in control cells. It might be informative to perform the same experiments upon nigericin treatment to investigate the ability of NLRP3 to recognize activating signals. It has been reported that PI4P increases on Golgi and endosomes upon NG treatment. Hence, all the differences between the domains may be lost or preserved. In parallel, also the timing of such recruitment upon nigericin treatment (early or late event) may be informative for the dynamics of the process and of the contribution of the single protein domains.

This is an interesting point which we thank the reviewer for highlighting. However, we think that each domain on its own is not capable of responding to nigericin as shown by the effect of mutations in helix115-125 or the PB region in the full-length NLRP3 protein. NLRP3HF, which still contains a functional PB region, isn’t capable of responding to nigericin in the same way as wild type NLRP3 (Supplementary Figure 6C-D). Similarly, mutations in the PB region of full length NLRP3 that leave helix115-125 intact show that helix115-125 is not sufficient to allow enhanced recruitment of NLRP3 to Golgi membranes after nigericin treatment (Supplementary Figure 9A). We speculate that helix115-125, the PB region and the LRR domain all need to be present to provide maximum affinity of NLRP3 for the Golgi prior to encounter with and S-acylation by ZDHHC3/7. Mutation or loss of any one of the PB region, helix115-125 or the LRR lowers NLRP3 membrane affinity, which is reflected by reduced levels of NLRP3 captured on the Golgi by S-acylation at steady state and in response to nigericin.

(5) As noted above for the chemical inhibitors (1) the authors should check the impact of altering the balance between acyl transferase and de-acylases on the Golgi organization and PI4P levels. What is the effect of overexpressing PATs on Golgi functions?

We have checked the effect of APT2 overexpression on Golgi morphology and can show that it has no noticeable effect, ruling out an impact of APT on Golgi integrity as the reason for loss of NLRP3 from the Golgi in the presence of overexpressed APT2. We have included these images as Supplementary Figure 11H-J.

It is plausible that the effects of ZDHHC3 or ZDHHC7 on enhanced recruitment of NLRP3 to the Golgi may be via an effect on PI4P levels since, as mentioned above, both enzymes are involved in recruitment of PI4K2A to the Golgi and have previously been shown to enhance levels of PI4K2A and PI4P on the Golgi when overexpressed (Kutchukian et al., 2021). However, NLRP3 mutants with most of the charge removed from the PB region, which are presumably unable to interact with PI4P or other negatively charged lipids, are still capable of being recruited to the Golgi by excess ZDHHC3. This would suggest that the effect of overexpressed ZDHHC3 on NLRP3 is largely independent of changes in PI4P levels on the Golgi and instead driven by helix115-125 and S-acylation at Cys-130. The latter point is supported by the observation that NLRP3HF and NLRP3Cys130 are insensitive to ZDHHC3 overexpression.

At the levels of HA-ZDHHC3 used in our experiments with NLRP3 (200ng pEF-Bos-HAZDHHC3 / c.a. 180,000 cells) we don’t see any adverse effect on Golgi morphology (Author response image 1), although it has been noted previously by others that higher levels of ZDHHC3 can have an impact on TGN46 (Ernst et al., 2018). ZDHHC3 overexpression surprisingly has no adverse effects on Golgi function and in fact enhances secretion from the Golgi (Ernst et al., 2018).

**Author response image 1. sa2fig1:** Overexpression of HA-ZDHHC3 does not impact Golgi morphology. A) Representative confocal micrographs of HeLaM cells transfected with 200 ng HA-ZDHHC3 fixed and stained with antibodies to STX5 or TGN46. Scale bars = 10 µm.

**Reviewer #2 (Public Review):**
Summary:This paper examines the recruitment of the inflammasome seeding pattern recognition receptor NLRP3 to the Golgi. Previously, electrostatic interactions between the polybasic region of NLRP3 and negatively charged lipids were implicated in membrane association. The current study reports that reversible S-acylation of the conserved Cys-130 residue, in conjunction with upstream hydrophobic residues plus the polybasic region, act together to promote Golgi localization of NLRP3, although additional parts of the protein are needed for full Golgi localization. Treatment with the bacterial ionophore nigericin inhibits membrane traffic and prevents Golgi-associated thioesterases from removing the acyl chain, causing NLRP3 to become immobilized at the Golgi. This mechanism is put forth as an explanation for how NLRP3 is activated in response to nigericin.Strengths:The experiments are generally well presented. It seems likely that Cys-130 does indeed play a previously unappreciated role in the membrane association of NLRP3.Weaknesses:The interpretations about the effects of nigericin are less convincing. Specific comments follow.(1) The experiments of Figure 4 bring into question whether Cys-130 is S-acylated. For Cys130, S-acylation was seen only upon expression of a severely truncated piece of the protein in conjunction with overexpression of ZDHHC3. How do the authors reconcile this result with the rest of the story?

Providing direct evidence of S-acylation at Cys-130 in the full-length protein proved difficult. We attempted to detect S-acylation of this residue by mass spectrometry. However, the presence of the PB region and multiple lysines / arginines directly after Cys-130 made this approach technically challenging and we were unable to convincingly detect S-acylation at Cys-130 by M/S. However, Cys-130 is clearly important for membrane recruitment as its mutation abolishes the localisation of NLRP3 to the Golgi. It is feasible that it is the hydrophobic nature of the cysteine residue itself which supports localisation to the Golgi, rather than S-acylation of Cys-130. A similar role for cysteine residues present in SNAP-25 has been reported (Greaves et al., 2009). However, the rest of our data are consistent with Cys-130 in NLRP3 being S-acylated. We also refer to another recently published study which provides additional biochemical evidence that mutation of Cys-130 impacts the overall levels of NLRP3 S-acylation (Yu et al., 2024).

(2) Nigericin seems to cause fragmentation and vesiculation of the Golgi. That effect complicates the interpretations. For example, the FRAP experiment of Figure 5 is problematic because the authors neglected to show that the FRAP recovery kinetics of nonacylated resident Golgi proteins are unaffected by nigericin. Similarly, the colocalization analysis in Figure 6 is less than persuasive when considering that nigericin significantly alters Golgi structure and could indirectly affect colocalization.

We agree that it is likely that the behaviour of other Golgi resident proteins are altered by nigericin. This is in line with a recent proteomics study showing that nigericin alters the amount of Golgi resident proteins associated with the Golgi (Hollingsworth et al., 2024) and other work demonstrating that changes in organelle pH can influence the membrane on / off rates of Rab GTPases (Maxson et al., 2023). However, Golgi levels of other peripheral membrane proteins

that associate with the Golgi through S-acylation, such as N-Ras, appear unaltered (Author response image 2.), indicating a degree of selectivity in the proteins affected. Our main point here is that NLRP3 is amongst those proteins whose behaviour on the Golgi is sensitive to nigericin and that this change in behaviour may be important to the NLRP3 activation process, although this requires further investigation and will form the basis of future studies.

The reduction in co-localisation between NLRP3 and APT2, due to alterations in Golgi organisation and trafficking, was the point we were trying to make with this figure, and we apologise if this was not clear. We think that the changes in Golgi structure and function caused by nigericin potentially affect the ability of APT2 to encounter NLRP3 and de-acylate it. We have added a new paragraph to the results section to hopefully explain this more clearly. We recognise that our results supporting this hypothesis are at present limited and we have toned down the language used in the results section to reflect the nature of these findings..

**Author response image 2. sa2fig2:** S-acylated peripheral membrane proteins show differential sensitivity to nigericin. (A) Representative confocal micrographs of HeLaM cells coexpressing GFP-NRas and an untagged NLRP3 construct. Cells were left untreated or treated with 10 µM nigericin for 1 hour prior to fixation. Scale bars = 10 µm. (B) Quantification of GFP-NRas or NLRP3 signal in the perinuclear region of cells treated with or without nigericin

Recommendations for the authors:

**Reviewer #2 (Recommendations For The Authors):**

(1) Does overnight 2-BP treatment potentially have indirect effects that could prevent NLRP3 recruitment? It would be useful here to show some sort of control confirming that the cells are not broadly perturbed.

Please see our response to point (2) raised by reviewer #1 which is along similar lines.

(2) In Figure 5, "Veh" presumably is short for "Vehicle". This term should be defined in the legend.

We have now corrected this.

References

Ernst, A.M., S.A. Syed, O. Zaki, F. Bottanelli, H. Zheng, M. Hacke, Z. Xi, F. Rivera-Molina, M. Graham, A.A. Rebane, P. Bjorkholm, D. Baddeley, D. Toomre, F. Pincet, and J.E. Rothman. 2018. SPalmitoylation Sorts Membrane Cargo for Anterograde Transport in the Golgi. Dev Cell. 47:479-493 e477.

Greaves, J., G.R. Prescott, Y. Fukata, M. Fukata, C. Salaun, and L.H. Chamberlain. 2009. The hydrophobic cysteine-rich domain of SNAP25 couples with downstream residues to mediate membrane interactions and recognition by DHHC palmitoyl transferases. Mol Biol Cell. 20:1845-1854.

Hollingsworth, L.R., P. Veeraraghavan, J.A. Paulo, J.W. Harper, and I. Rauch. 2024. Spatiotemporal proteomic profiling of cellular responses to NLRP3 agonists. bioRxiv.

Kutchukian, C., O. Vivas, M. Casas, J.G. Jones, S.A. Tiscione, S. Simo, D.S. Ory, R.E. Dixon, and E.J. Dickson. 2021. NPC1 regulates the distribution of phosphatidylinositol 4-kinases at Golgi and lysosomal membranes. EMBO J. 40:e105990.

Mateo-Tórtola, M., I.V. Hochheiser, J. Grga, J.S. Mueller, M. Geyer, A.N.R. Weber, and A. TapiaAbellán. 2023. Non-decameric NLRP3 forms an MTOC-independent inflammasome. bioRxiv:2023.2007.2007.548075.

Maxson, M.E., K.K. Huynh, and S. Grinstein. 2023. Endocytosis is regulated through the pHdependent phosphorylation of Rab GTPases by Parkinson’s kinase LRRK2. bioRxiv:2023.2002.2015.528749.

Yu, T., D. Hou, J. Zhao, X. Lu, W.K. Greentree, Q. Zhao, M. Yang, D.G. Conde, M.E. Linder, and H. Lin. 2024. NLRP3 Cys126 palmitoylation by ZDHHC7 promotes inflammasome activation. Cell Rep. 43:114070.